# Nanoscale silicate melt textures determine volcanic ash surface chemistry

Adrian J. Hornby [1,2,8] ✉, Paul M. Ayris[2,8], David E. Damby[3], Spyridon Diplas[4], Julia Eychenne[5,6], Jackie E. Kendrick[2], Corrado Cimarelli [2], Ulrich Kueppers [2], Bettina Scheu[2], James E. P. Utley[7] & Donald B. Dingwell [2]

Explosive volcanic eruptions produce vast quantities of silicate ash, whose surfaces are subsequently altered during atmospheric transit. These altered surfaces mediate environmental interactions, including atmospheric ice nucleation, and toxic effects in biota. A lack of knowledge of the initial, pre-altered ash surface has required previous studies to assume that the ash surface composition created during magmatic fragmentation is equivalent to the bulk particle assemblage. Here we examine ash particles generated by controlled fragmentation of andesite and find that fragmentation generates ash particles with substantial differences in surface chemistry. We attribute this disparity to observations of nanoscale melt heterogeneities, in which Fe-rich nanophases in the magmatic melt deflect and blunt fractures, thereby focusing fracture propagation within aureoles of single-phase melt formed during diffusion-limited growth of crystals. In this manner, we argue that commonly observed pre-eruptive microtextures caused by disequilibrium crystallisation and/or melt unmixing can modify fracture propagation and generate primary discrepancies in ash surface chemistry, an essential consideration for understanding the cascading consequences of reactive ash surfaces in various environments.

The chemistry of ash particle surfaces generated by volcanic eruptions is an important but poorly understood mediator of many volcanic, atmospheric, and environmental processes. In an eruption plume, adsorption of hot volcanic gases ($SO_2$, HCl, HF, etc.) and reaction with condensing acid droplets extract elements from the ash surface to form a range of surface compounds, commonly salts[1]. Secondary surface mineralisation and in-plume modification of ash surface chemistry can enhance ash aggregation[2] and modify the ice nucleation efficiency[3] and charge-carrying capacity of ash surfaces, consequently affecting volcanic lightning and fallout processes[4].

The potential for ash depositing into terrestrial, marine and anthropogenic settings to act as an environmental[5,6], commercial[7] and health[8] hazard has driven strong interest in ash surface chemistry at a nanoscale. Previous studies[9,10] have invoked alteration by various in-plume and post-eruptive processes to explain differences between the nanoscale surface and bulk chemistry of the ash particles. However, no previous study has considered whether such discrepancies could derive from an a priori heterogeneity created from the microtextures and fragmentation mode(s) of crystal-bearing magmas during fragmentation[11,12].

[1]Department of Earth and Atmospheric Sciences, Cornell University, Ithaca, NY, USA. [2]Department of Earth and Environmental Science, Ludwig-Maximilians-Universtität (LMU), München, Germany. [3]U.S. Geological Survey, Volcano Science Center, Menlo Park, CA, USA. [4]Material Physics Oslo, SINTEF Industry, Oslo, Norway. [5]Université Clermont Auvergne, CNRS, IRD, OPGC, Laboratoire Magmas et Volcans, F-63000 Clermont-Ferrand, France. [6]Université Clermont Auvergne, CNRS, INSERM, Institut de Génétique Reproduction et Développement, F-63000 Clermont-Ferrand, France. [7]Department of Earth, Ocean & Ecological Sciences, University of Liverpool, Liverpool, UK. [8]These authors contributed equally: Adrian J. Hornby, Paul M. Ayris. ✉e-mail: ahornby@cornell.edu

Most magmas are multiphase systems bearing crystal phases that vary in fracture toughness (the resistance to crack propagation[13]). In addition, disequilibrium chemical and physical textures commonly develop in silicate melts during episodes of shallow storage, magma mixing, and rapid ascent prior to eruption[14,15], causing rheological changes that often promote violent explosive activity[16–18]. Fragmentation of multiphase magmas during explosive eruptions likely entails phase- and texture-sensitive fracture propagation[19], resulting in primary differentiation of particle surfaces detectable at the micronscale[20].

Here, we highlight the influence of textural and magmatic drivers on the nanoscale ash surface of experimentally fragmented volcanic rock cores. The volcanic materials were generated in the VEI 3 eruption of Tungurahua volcano, Ecuador[21] on August 16–17th, 2006. We do not argue for universal applicability of the specific fracture-focusing model outlined here to all volcanic surfaces but rather use it to underpin an essential argument. If chemically discrepant ash surfaces are created from combinations of microtextural and fragmentation conditions in one situation, then there may be an array of variably discrepant surfaces created by interactions between microtexture and fracture propagation across the spectrum of ash-forming eruptive events. Any future studies of ash surface-mediated reactions may be compromised in their utility as long as this subject area remains unexplored.

## Results and discussion

### Fragmentation experiments

Experimental samples in this study were 25 mm diameter cores, drilled from a juvenile, scoriaceous bomb[22] of 5 cm diameter (Fig. 1). The bomb was collected from primary deposits of pyroclastic density currents (PDCs) 4 km from the vent. This sample was emplaced by PDCs generated by the intermittent collapse of the ballistic and fallout material (bombs, scoriae, ash), accumulated on the crater rim by the fountaining activity which occurred during the August 16–17th eruption[23]. The textures and bulk composition of all core samples were similar, however the ash fallout samples showed bimodal size distributions[24] and heterogeneous petrographic textures attributed to contributions from the plume-fed ash cloud and from co-PDC elutriated ash (see detailed description in Methods). After fragmentation experiments of cores in a shock tube apparatus (see Methods), we separated recovered particles into two size fractions: a 63–90 μm-sized fraction, which was reserved for surface and bulk analysis, and particles >1 mm diameter, which we subsequently crushed and sieved to the same 63–90 μm size-fraction in order to generate material that had undergone the same experimental pressure and temperature conditions but a different fragmentation process. Hereafter, we refer to these two sets of samples as fragmented and crushed, respectively.

### Multi-scale measurements of surface chemistry

Comparison of the nanoscale (<10 nm depth resolution) surface composition measured by X-ray photoelectron spectroscopy (XPS) against the bulk sample composition by X-ray fluorescence (XRF) revealed significant discrepancies (see Methods and Supplementary Table 1, 2). Compared to past studies of natural ash samples, the variations in elemental enrichment or depletions at the surface occur over a broadly similar range (approximately, ratios between 0.1 and 3)[9,10]. However, within that range, there are some remarkable features: Fig. 2a shows that, for all experimental sample surfaces, Mg is strongly depleted (ratios between 0.14 and 0.44); Fe is moderately depleted, with ratios from 0.55 to 0.9); and Na, K and Al are typically enriched (ranges of 0.95–2; 1.05–1.25; 0.95–1.2, respectively). These general trends are observed for all experimental conditions and fragmentation mechanisms, although, at a finer-scale, we observe systematically lower Mg and Na and higher Si in the nm-scale ash surfaces produced by shock tube experiments compared to those produced by crushing (Fig. 2a).

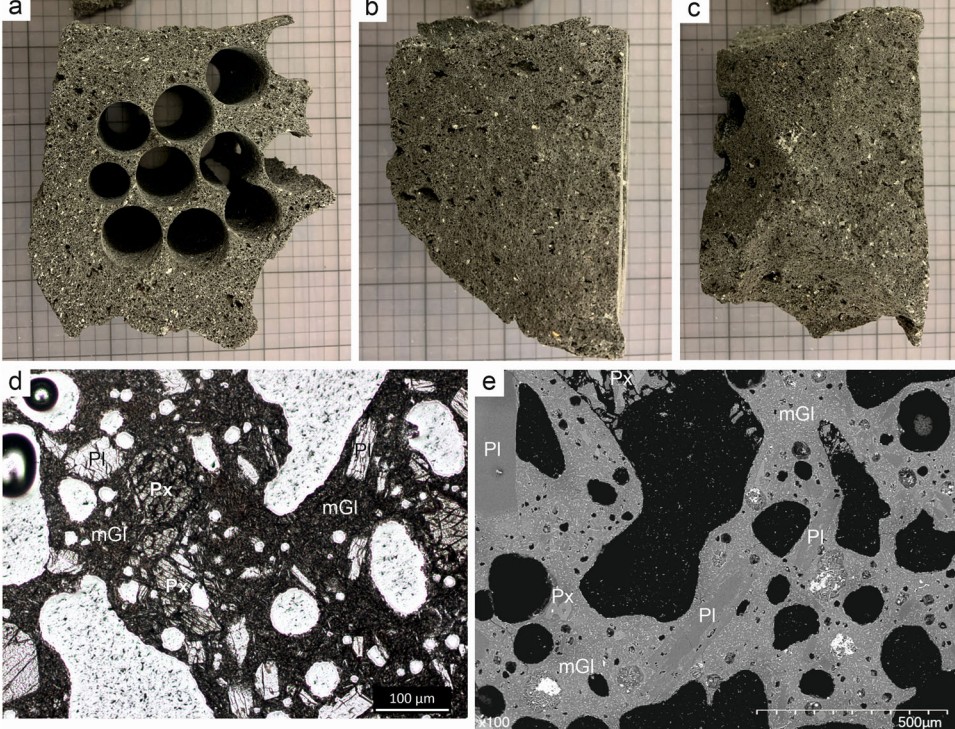

**Fig. 1 | Macro-to-micro textures of starting materials. a** Photo of the scoriaceous bomb collected from PDC deposits of the 16–17th August 2006 eruption of Tungurahua on a 1 cm grid. The top surface was cut flat, and cores were drilled perpendicular to the cut surface. **b**, **c** Orthogonal side views of the starting block, showing the pore texture along the axis of the drilled cores. **d** Microphotograph in transmitted light and (**e**) SEM-BSE image, showing pore structure (in white in (**b**) and in black in (**c**)), microlite rich matrix glass (mGl) and crystals of plagioclase (Pl) and pyroxene (Px).

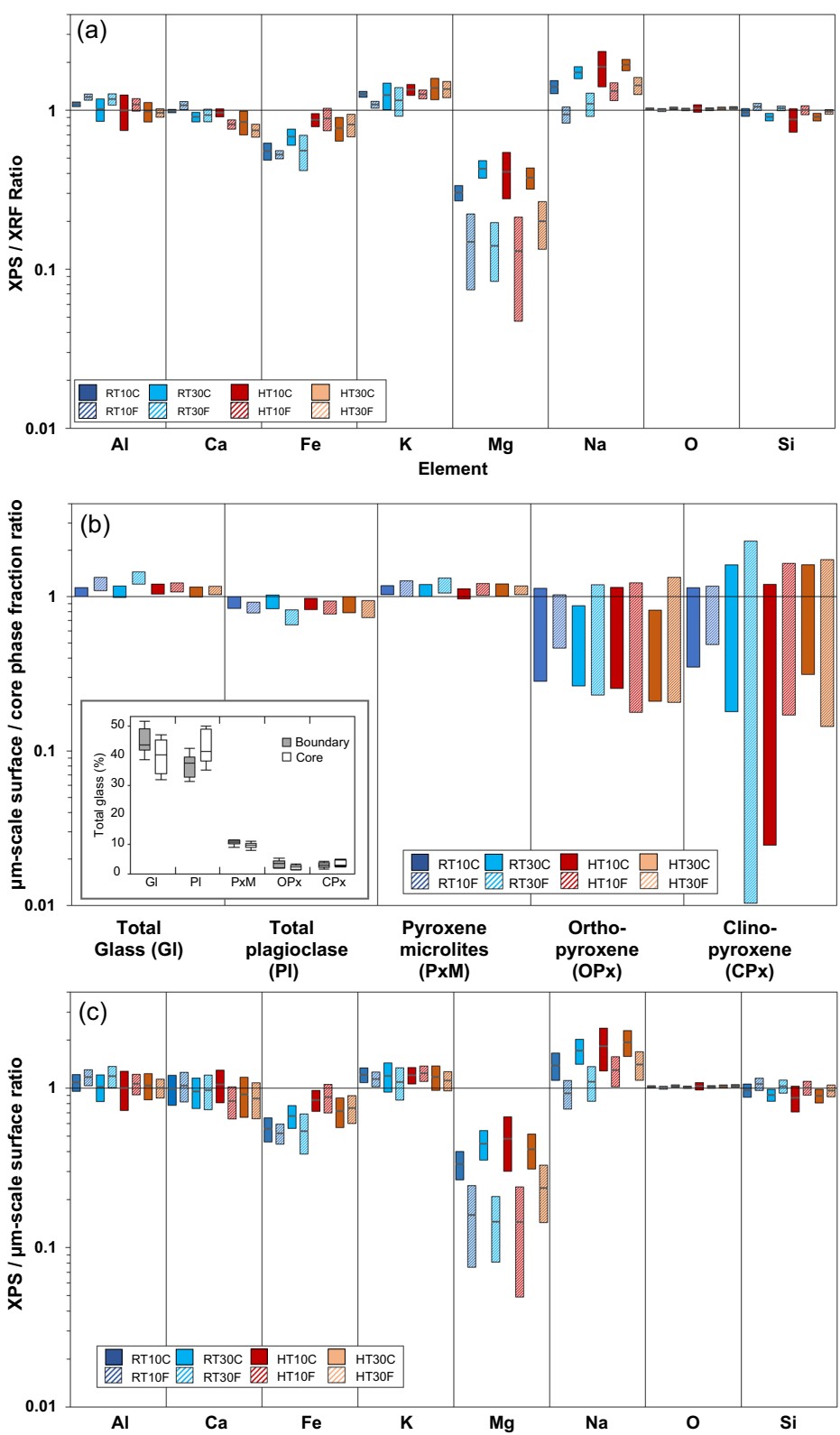

**Fig. 2 | Surface and bulk composition of the fragmented clasts. a** The quotient of the elemental abundance at the nanoscale surface of experimental clasts measured by XPS with the average bulk composition measured by XRF. **b** Relative variation in the major phases between the bulk and the micron-scale surface of particles as measured by QEMSCAN. Positive values indicate phase enrichment at the surface. The average phase composition from all samples is shown in the inset. **c** The quotient of the nanoscale surface chemistry measured by XPS and the microscale surface chemistry calculated from EPMA and QEMSCAN data (calculated μm-scale surface). The coloured bars show ±2× standard error for all panels and mean absolute error is shown in the inset to (**b**). In the legend, RT and HT indicate room temperature or high temperature (850 °C), respectively, 10 and 30 refer to confining pressure (in MPa) in shock tube experiments, and the suffix letter refers to samples produced by crushing (C) or decompression fragmentation (F).

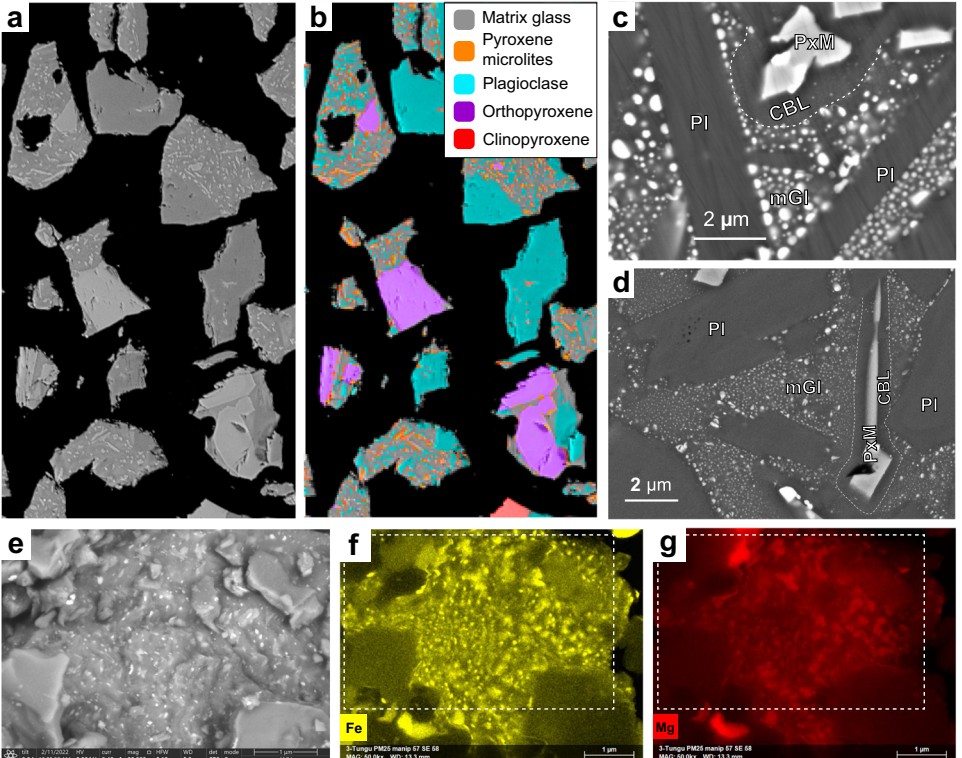

**Fig. 3 | Textural and petrological characterisation of fragmented clasts.** SEM-BSE image (**a**) and overlain QEMSCAN phase map (**b**) for polished clasts from a crushed block, with phases given in the key. **c** SEM-BSE image showing high-magnification matrix texture, including euhedral plagioclase feldspar (Pl) and pyroxene microlites (PxM) hosted in a glass matrix (mGl) containing numerous rounded, nm-scale bright features that may represent Fe-rich nanolites or immiscible globules. This phase is absent in a ~1 μm thick zone surrounding pyroxene microlites marked by a dashed line; the zone has decreasing BSE intensity toward the crystal surface, suggesting a compositional boundary-layer depleted in Fe relative to the matrix glass. **d** SEM-BSE image of the matrix in a natural volcanic ash particle from the August 2006 eruption of Tungurahua, showing the same nanoscale textural features described in (**c**). **e** SEM-SE image of the unpolished surface of an ash grain from the August 2006 eruption of Tungurahua showing bright nanoscale speckles. SEM-EDX analysis (**f**, **g**) with dashed box showing area in (**e**) shows that the speckles are enriched in Fe (**f**) and Mg (**g**).

To investigate whether the observed variations in surface chemistry can be explained by any partitioning of phases at particle surfaces discernible at the micron-scale, we performed QEMSCAN analysis, an automated mineralogy procedure using energy dispersive X-ray (EDX) spectroscopy[25]. The QEMSCAN analysis produced phase maps limited to the pixel resolution of 1.8 microns and depth resolution of 1–2 microns, which have been uploaded to a public repository (see Data Availability). Phase analysis showed plagioclase feldspar as the dominant crystalline phase in a glassy matrix also hosting pyroxene and Fe-Ti oxide microlite populations (Fig. 1b, Supplementary Table 3, Supplementary Table 1). We found that the microscale particle boundaries of the particles from all experimental samples are slightly enriched (by 10–15%) in pyroxene microlites and matrix glass and depleted in plagioclase compared to the bulk (Fig. 2b). No consistent trends were observed with experimental temperature or pressure and differences between crush and shock tube experiments were within error.

To determine whether the measured variations in microscale surface mineralogy could account for the disparate nanoscale surface chemistry, we combined phase abundance and chemical composition data measured by electron probe microanalysis (EPMA) to calculate hypothetical material compositions in the bulk and near-surface region (Supplementary Table 4). EPMA data show a bimodal population of high-Mg pyroxene microlites, the majority pigeonite with 10–20% enstatite, both with Mg/Mg + Fe ~0.7 (Supplementary Table 2). Glass compositional measurements were hampered by smearing effects due to the defocused beam diameter (10 μm) and covered a wide chemical range. Accordingly, we used iterative goal-seeking to calculate a single, bulk matrix glass composition which could combine with the measured data for crystal phases to yield a material composition equivalent to the bulk XRF data (see Methods). Although the calculated matrix composition (which models the melt composition including nanoscale phases that are below the resolution of QEMSCAN mapping) is broadly compatible with glass measurements obtained by EPMA (Supplementary Table 5, Supplementary Table 3), variations in Ca, Fe and Mg are noted, likely due to the variable contribution of unavoidable micro-to-nanoscale features in EPMA measurements. With these data and calculated values, we show that the microscale mineral variation in the near-surface region cannot account for variability in the nanoscale surface chemistry observed by XPS (Fig. 2c, Supplementary Table 6, Supplementary Fig. 1). Indeed, notwithstanding that use of the calculated Mg concentration (0.22 at.%) instead of defocused EPMA measurements (0.6–1.16 at.% Mg) of the matrix glass reduces the discrepancy between microscale and nanoscale surface compositions, we find nanoscale surfaces depleted in Mg by 2–20× across all samples (Fig. 2c). Therefore, differences in the microscale surface mineral abundance between shock tube and crushed samples measured by QEMSCAN cannot be directly linked to the nanoscale surface composition but indicate a lower magnitude effect (we elaborate on these variations in Supplementary Discussion 1).

## Disequilibrium micro-to-nanotextures

The QEMSCAN phase maps and SEM images at low magnification do not show fine-scale properties of the matrix (Fig. 3a, b, Supplementary Fig. 2). In search of evidence that could account for the nanoscale surface chemistry, we examined the experimental materials using high-resolution SEM backscattered electron (BSE) imaging. We

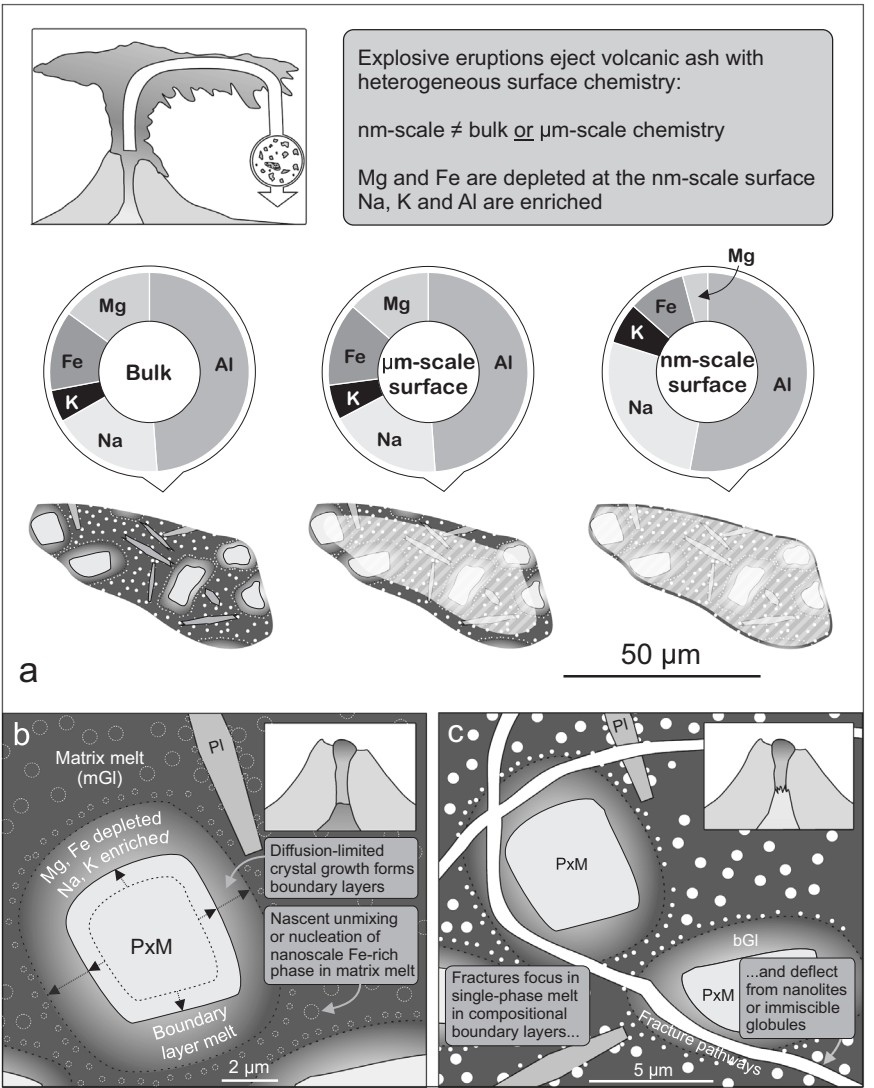

**Fig. 4 | Model for scale-dependent volcanic ash surface chemistry via fragmentation through a heterogenous melt. a** Volcanic ash particle surfaces produced by magma fragmentation have a similar composition to the bulk at the micron-scale but can show significant variations from the bulk at the nanoscale. Chemical analysis results from experimentally fragmented Tungurahua ash are shown for selected elements for the non-shaded volumes. **b** Diffusion-limited growth of mafic microlites during pre-eruptive magma mixing, ascent and storage causes boundary-layer formation within the melt. Subsequent nucleation and/or unmixing in the matrix melt produces nanoscale Fe-rich phases whose size and number density is sensitive to the local melt chemistry; they reduce in size toward the outer extreme of the Fe- and Mg-depleted boundary-layer melt and become absent within ca. 1 um of the mafic crystal boundary. All phase labels are the same as in Fig. 2. **c** The interstitial melt bears nanoscale Fe-rich phases and contains Mg + Fe-depleted boundary-layer melt around mafic crystals. During magma fragmentation, fractures preferentially propagate through the single-phase glass in boundary layers (<1 µm from mafic crystal faces) and may deflect around the Fe-rich nanolites or immiscible globules in the unmixed melt.

observed the presence of an abundant nm-scale phase or phases with high BSE intensity (Fig. 3c and Supplementary Figs. 2–7) in the matrix glass. Larger examples (>50 nm) have smooth, rounded shapes, whose populations (1) coarsen towards plagioclase crystals and (2) become smaller and finally disappear in a micron-wide zone around mafic crystals (Fig. 3c). The phase(s) represent either Fe-rich nanolites, a common feature documented in detail[18,26,27] and/or immiscible melt globules[28,29], as suggested by the subspherical form. The nanoscale Fe-rich phases are ubiquitous in the matrix glass, except in a narrow (~1 µm) band surrounding mafic crystals (the most abundant of which are pyroxene microlites), which is also distinguished by progressively decreasing BSE intensity toward the crystal edges. SEM-EDX analysis of the matrix glass chemistry from the edge of pyroxene microlites outward into the matrix glass shows relative depletions of Fe and Mg (extending for ~2 µm) while Na and K are relatively enriched in this zone (Supplementary Fig. 3). We interpret the variations in element concentration and decreasing SEM-BSE intensity towards pyroxene microlites as evidence for diffusion-limited crystal growth preserving compositional boundary layers within the magmatic melt[15,28,30]. No difference is found in the matrix textures or compositional boundary layers between room temperature and 850 °C experiments, indicating that the original textures were not substantially altered by heating up to 150 °C above the estimated glass transition temperature for our experimental duration (Supplementary Discussion 2, Supplementary Fig. 4). We can exclude their formation by crystal dissolution since Fe concentration (and consequently BSE intensity) would steadily decrease away from, rather than towards, the mafic crystal boundaries (note the differences in Fe concentration between pyroxenes and glass; Supplementary Tables 2, 3).

To determine whether these textures were unique to our experimental bomb, we compared them with natural pyroclasts in the same 63–90 µm size range collected from the same phase of the eruption.

We found that 15–20% of the natural pyroclasts had matrix textures that matched (and are usually indistinguishable from) our experimental samples (Fig. 3c, d and Supplementary Figs. 5, 6); these same textures can be seen in a previous study, however without detailed characterisation[31]. These natural samples have been interpreted as having been formed in PDCs generated by repeated collapse of summit and flank ramparts built during prolonged fire fountaining activity prior to the paroxysmal phase of the eruption (see Methods for further details). Neither the nanoscale Fe-rich phase nor the compositional boundary layers are discernible by the microscale QEMSCAN mapping or from low-magnification SEM-BSE micrographs[31] (Fig. 3a, b, Supplementary Fig. 2).

To gain direct insight into the effect of these nanoscale textures on crack propagation, and therefore generation of the nanoscale compositional anomalies that we observe by XPS (Fig. 2a, c), we conducted a survey of incomplete fracture paths passing through particles in the experimental samples. We observe that fractures are often thinner and more frequently terminate within the nanotilized matrix glass than in crystal phases. We also note a tendency for fractures to deviate around the edges of pyroxene microlites, rather than passing through them (Supplementary Fig. 7).

## Fracture-focusing model

On the basis of our high-resolution investigations, we offer here a conceptual model to account for the discrepancy between the nanoscale surface chemistry and the bulk or microscale surface chemistry in our experimental materials (Fig. 4a). During fragmentation, we propose that fracture deviation, blunting or branching around nanoscale phases in the matrix glass, and consequently the preferential propagation of fractures through single-phase Mg- and Fe-depleted glass in compositional boundary layers, forms nanoscale particle surfaces with distinctive Mg- and Fe-depleted and alkali-enriched compositions.

This model is supported by the glass and ceramics literature, where recent studies have shown that seeding of nanolites or immiscible globules[32] in a silicate glass increases fracture toughness via crack tip bridging, deflection and blunting[33], capable of producing the toughest inorganic glass ceramic known to date[34]. Therefore, the presence of a nanoscale phase in natural magmatic melts may effectively inhibit particle-bounding fractures in a similar fashion during eruptive fragmentation. By deduction, we expect particle-forming fractures to preferentially propagate through regions of the matrix where the nanoscale phase is absent or lower in abundance, particularly within the nanophase-free melt in compositional boundary layers surrounding mafic crystals in our natural samples (Fig. 4b). This leads to an over-representation of the chemistry of the boundary-layer glass at the surface of the particles. Supplementary Fig. 3 demonstrates that the compositional boundary layers are depleted in Fe and Mg and enriched in Na and K. Thus, fracture propagation modulated by the presence of the nanoscale phase would create a surface with the same compositional trends evidenced in our XPS data (Fig. 4a). While it is likely that the compositional gradients in the boundary-layer melt also influence fracture propagation, we interpret this to be minor relative to the effect of the nanoscale Fe-rich nanolites and/or globules (see Supplementary Discussion 2).

## Natural evidence and model scope

The effects of nanoscale phases[17] and dynamic crystallisation are both understudied processes in magma textural evolution, and explosive deposits bearing evidence for such features have been infrequently documented[35,36] until recent years[37–39]. At Tungurahua, geophysical and petrological data suggest that recharge and mixing of mafic magma caused magma to ascend from a reservoir at 8–10 km to 2–4 km depth[21,40] in the months leading up to a large eruption on July 14 and the subsequent VEI 3 explosive eruption on August 16–17, 2006.

Magma decompression, cooling and degassing accompanying this month-long period of shallow pre-eruptive magma ascent and storage is likely to have been accompanied by rapid, diffusion-limited crystal growth, creating compositional boundary layers in adjacent melt and nano-crystallisation of the groundmass glass (Fig. 4c). The observation of these features in the proximal pyroclastic fall deposits at Tungurahua (see Fig. 3, Supplementary Figs. 5, 6 and Methods) demonstrates that these textures formed in-situ and were disrupted during fragmentation, rather than forming via reheating or rapid cooling, processes that can generate similar textures[41,42]. Our observations and inference of interaction between nanoscale textures and fractures suggest that their development may affect the timing and efficiency of fragmentation processes (see Supplementary Discussion 3) and plays an important role in determining the ash surface chemistry.

Here, we define a model of ash surface creation modulated by rapid pre-eruptive growth of mafic crystals and the presence or development of a nanoscale Fe-rich phase in the matrix glass. These features have been documented in explosive products from mafic (Stromboli[37] and Mt Etna including the 122 BC Plinian eruption[18]) through silicic (Havre[43]) explosive eruptions. The conditions for formation of the set of melt nanoheterogeneities that we document here (described in detail in Supplementary Discussion 4) are most applicable to relatively oxidised arc environments, particularly those with higher Fe and alkali content, and for magma systems perched close to a critical threshold where mafic recharge may cause disequilibrium crystallization or unmixing and lead to sharp viscosity increases and explosive eruption[17,18].

However, a broader range of heterogeneous melt textures generated during the disequilibrium conditions that commonly accompany or trigger magma ascent, shallow storage and eruption are likely to affect magma fragmentation paths and subsequent ash surface chemistry. The rapid decompression experiments in this study reproduce natural fragmentation triggered by dome or edifice collapse or unloading events (rapid decompression) including plugged Vulcanian-style eruptions. Such eruptive events and phenomena are frequently observed at explosive volcanoes with intermediate magma compositions in subduction settings. These are the most frequently erupting volcanoes[44], and hence are the dominant sources of volcanic ash on a global scale (e.g., >70 wt.% of ash + PDC deposits from VEI 1–5 eruptions in subduction settings during the last 40 years[45]). Although we cannot claim that the rapid decompression experiments well-reproduce the fragmentation process of the erupted material during the sub-Plinian phase of the Tungurahua 2006 eruption, the crushed experimental samples better simulate secondary fragmentation by collisions between pyroclasts in PDCs[22], which is the mechanism that has been proposed to fragment the natural ash bearing the nanotextures described here (see Methods, Fig. 2e–g, Supplementary Fig. 5).

Within these limits, our experiments suggest that the sensitivity to nanotextures would apply to all but the most energetic fragmentation modes, for example, phreatomagmatic and Plinian strain-induced fragmentation[46], where further investigation is required. Excluding such high-energy eruption modes, we can conclude that the surface chemistry of ash from eruptive events, and by extension, the reactivity of that surface, may differ from that implied by a micro- or bulk scale analysis. For example, emissions of Ca- and Fe-rich ash may be inferred from bulk chemistry to be a sink for volcanogenic $SO_2$[1,47] and a contributor of bioavailable Fe to surface waters; however, a nanoscale surface enriched in alkalis and silica would likely be of little relevance to either phenomena.

We emphasise that the model outlined for the Tungurahua materials in this study is not to be universally applied. However, the foundational principle put forward by this study, that nanoscale surface chemistry of ash surfaces is likely strongly influenced by the textural, magmatic and fragmentation conditions, should be considered

in all cases. Consequently, we advocate for detailed observation and measurements of surface and near-surface features, together with targeted experimental and analytical work with a focus on discerning the influence of textural and fragmentation controls on those surfaces. Such steps are to accurately (re)assess the host of ash surface-mediated reactions and impacts to the atmosphere and depositional environments.

## Methods

### Natural sample collection and generation

The starting material for the fragmentation experiments is a porous (25–35 vol. %) andesitic (58 wt.% $SiO_2$) bomb sampled from deposits of the August 16–17 2006 VEI 3 eruption of Tungurahua volcano, Ecuador[48,49]. The bomb was taken from the top of primary PDC deposits that were emplaced in the early morning of August 17 (local time) during the paroxysmal (Phase III[23]) stage of the eruption, which produced a 16 km high eruptive plume. Two natural samples were collected from the pyroclastic fall deposit at 7.9 km (sample F2) and 17.7 km (sample F11) from the eruptive vent. Detailed investigation of the different pyroclastic deposits from the 2006 eruption indicates contrasting eruptive history (and therefore microtexture) between the material deposited from the main eruptive plume and the material emplaced by the PDCs and co-PDC plumes. During Phase I-II of the eruption, Strombolian activity and intense fire fountaining built unstable scoria cones, ramparts and scarps around the summit and the north and west flanks[23,49]. PDCs generated during Phase II and Phase III are thought to derive from successive collapse of these structures, therefore entraining previously erupted ballistic material, and not from column collapse of the eruptive plume during the paroxysmal phase. The ash fraction in the PDC deposits is thought to form by milling of the bigger grains during transport[22], with the ash finer than 90 μm preferentially lofted in co-PDC plumes. This co-PDC ash deposited concomitantly to the pyroclasts from the vent-derived plume formed during Phase III, and is hence preserved in the widespread fallout deposit which shows characteristic size distributions[24]. Componentry analysis indicates that the August 2006 eruption was triggered by a deep mafic magma reinjection[50]. The eruption dynamics and textural data[31] (see also Supplementary Fig. 6) suggest that a more viscous and crystalline magma which was stalling in the edifice following the previous eruptive phase in July 2006 erupted during the initial Phases I-II, while the paroxysmal plume-forming Phase III involved a rapidly ascending magma with low micro-crystallinity. Therefore, the bomb collected is representative of the magma involved in Phase I-II prior to the paroxysmal Phase III, comprising ~40 vol.% of the total erupted deposits[49]. The fraction of ash generated in such collapse-fed PDCs and found in the fallout deposit varies with proximity to the PDC deposits and the axis of dispersion for co-PDC ash clouds, however image analysis of the proximal deposits indicates 15–20% (mean = 19.3%, median = 16.5% - see Supplementary Fig. 6) of grains in the 63–90 micron size range have matrix glass microtextures indistinguishable to those in the studied bomb.

### Generation of fragmented and crushed samples

Cores of 25 mm diameter were drilled from the bomb described above. We conducted fragmentation experiments in a shock tube at Ludwig-Maximilians-Universität München according to the methods of Alidibirov and Dingwell[51] using argon gas to pressurise core samples in an autoclave to either 10 or 30 MPa, and under room temperature or heated to 850 °C using a split-cylinder furnace. Heated experiments were left at the target temperature for 1 h before fragmentation. The core sample and overlying void space was enclosed using a scored steel diaphragm (diaphragm thickness and score depth empirically determined to rupture at a known differential pressure, always <target pressure), separating it from and overlying smaller volume also sealed

with a scored diaphragm. Both volumes are pressurised simultaneously, with the upper chamber pressure at 50% final pressure in the lower chamber. Once final pressure is reached in the lower volume containing the core sample, fragmentation was triggered by a sudden increase in pressure in the upper chamber only, causing the uppermost diaphragm to fail followed by the near-instantaneous rupture of the lower diaphragm. Gas and particles are ejected into a 4 m high by 0.4 m diameter cylindrical tank at room pressure and temperature. After fragmentation experiments, we separated recovered particles into two size fractions: a 63–90 μm-sized fraction, which was reserved for surface and bulk analysis, and particles >1 mm diameter. These larger clasts were wrapped in paper and gently struck with a rock hammer and then sieved to the same 63–90 μm size-fraction to generate material that had undergone the same experimental pressure and temperature conditions but a different fragmentation process. We refer to these two sets of samples as fragmented and crushed samples, respectively.

### X-ray photoelectron spectroscopy

X-ray photoelectron spectroscopy (XPS) analysis was conducted on a Kratos XPS/UPS instrument with a monochromatic Al source (1486.1 eV) at the SINTEF/University of Oslo XPS lab. Sample particles with a 63–90 μm size range were placed in a Cu-based KRATOS sample holder for powders in sufficient amounts to fully cover the bottom of the cup-shaped sample holder. Each analysis consisted of an initial full-spectrum scan at low resolution, followed by high resolution analysis of the specific binding energy regions associated with Si (2$p$), Al (2$p$), Fe (2$p$), Na (1$s$), K (2$p$), Ca (2$p$), Mg (2$p$), C (1$s$) and O (1$s$) core level electrons. Acquired XPS data were fitted using CasaXPS version 2.3.9; spectra were normalised to adventitious carbon at 284.8 eV, and deconvoluted utilising linear baselines.

To reduce the influence of sample heterogeneity on data, five discrete XPS analyses were taken at different locations for each loose powder sample. The analytical area was approximately $300 \times 700$ μm with a depth resolution of 2–10 nm. We estimate that each analysis sampled at least six to ten particles, and the composite spectra of 5 separate analyses of the same sample reflects the contribution of at least thirty particles. Such analyses should not be considered as generally necessary for characterisation of volcanic ash, but in the current analysis, where interpretation of fine-scale surface features is necessary, the chance measurement of a large crystal in a single scan would impart a significant risk to the utility of the dataset. A detailed discussion of measures taken to ensure XPS data integrity are included as Supplementary Discussion 5.

### Electron probe microanalysis

EPMA was carried out on both crushed and fragmented clasts using a Cameca SX-100 system at Ludwig-Maximilians-Universität München. Samples were analysed for Na, Si, K, Ca, Fe, Mg, Al, P, Ti, and Mn using a defocused (10 μm) beam for glass analysis and a focused beam for crystal phase analysis. Measurements were performed at an accelerating voltage of 15 keV and a current of 5 nA. All EPMA data and instrument calibration details used in the study are included in Supplementary Data 1 (focused beam) and 2 (defocused beam).

### QEMSCAN analysis

All aliquots were analysed by QEMSCAN (Quantitative Evaluation of Minerals by scanning electron microscopy) at the University of Liverpool using a QEMSCAN Wellsite system. QEMSCAN is an automated mineralogy technique that uses a combination of scanning electron microscopy backscattered electron (SEM-BSE) intensity and energy dispersive X-ray spectroscopy (SEM-EDX) to map a geological sample on a polished mount[25]. The chemical elemental data and SEM-BSE intensity from each measurement point is used to assign a phase by constructing a customised species identification protocol (SIP). We

used a sub-2-micron step size to create phase maps of the particles in each sample. The groundmass glass composition measured by EPMA was included in the SIP to ensure an accurate match, and post-processing phase identification was optimised using EPMA, SEM-BSE and SEM-EDX data. In this analysis, solid-solution series were grouped under a single category (e.g., plagioclase feldspar) and all silica polymorphs were categorised as quartz. QEMSCAN outputs a particle-size-sorted false colour raster image where each pixel represents a measurement point, with a unique colour for each mineral or phase.

### SEM-EDX and BSE analysis

High-magnification SEM imaging and chemical analyses were conducted on a Hitachi SU5000 FE-SEM equipped with an Oxford Instruments X-max detector at Ludwig-Maximilians-Universität (LMU) München (SEM-BSE and SEM-EDX measurements), a TESCAN Mira3 FE-SEM at Cornell University (SEM-BSE) and a Helios 5 FE-SEM (Thermo Fisher Scientific) equipped with a 60 mm² annular Bruker FLATQUAD detector (SEM-SE and SEM-EDX measurements). Experimental and natural samples were polished on carbon-coated epoxy mounts or dispersed on carbon sticky tape for SE imaging. EDX measurements were typically made using 10–20 kV accelerating voltage at a working distance of 10–15 mm. Beam calibrations at LMU were made using a pure Cu K-series peak, and elemental calibrations were made using albite for Na, MgO for Mg, alumina for Al, silica for Si, KBr for K, wollastonite for Ca and pure metals for Ti and Fe. We used K-series X-ray peaks for Na, Mg, Al, Si, K, Ca, Ti and Fe to measure elemental abundances and calculated oxygen by stoichiometry. SEM-BSE images were collected at 10–20 kV and a working distance of 5–10 mm.

### Image analysis

8-bit colour TIFF images of QEMSCAN maps were used to create single-phase images by thresholding within ImageJ. Single-pixel particle boundary images in false colour were also produced in ImageJ and saved as single-phase TIFF files. All images used in the analysis have been uploaded into a public data repository (see Data Availability). The average distribution of phases in the ash samples was calculated using the Colour Counter plugin for ImageJ. Error in the mean was calculated by saving regions of interest in ImageJ and calculating the fraction of each phase within each ROI using the MultiMeasure function on single-phase image stacks. This was repeated on the particle boundary image stacks, using a minimum particle area of 68 µm² for all analyses. Separately, an ImageJ segmentation macro was used to measure the fraction of particle containing relatively low-Fe glass in natural ash samples. The QEMSCAN analysis and RockPie macros used can be found on https://github.com/hornbya/.

### XRF and micro-XRF analysis

Bulk XRF measurements were made of crushed experimental materials sieved to the same 63–90 µm fraction as all other measurements. Measurements were conducted using a Panalytical MagiXPro spectrometer at Johannes Gutenberg-Universität Mainz. The lab provides a commercial service and regularly calibrates the instrument using international standards. For major elements, samples were mixed with lithium tetraborate, melted in Pt crucibles and quenched to form glass tablets prior to measurement. Mass loss on ignition was within error. Due to small sample sizes for fragmented samples, XRF analysis was not possible. However, micro-XRF measurements were conducted for all 63–90 µm sieved samples using a Bruker S8 Tiger wavelength dispersive XRF analyser at Umeå University, Sweden. Sample masses between 0.13 and 0.21 g were analyzed according to the LAK E_200mg calibration method[52]. The calibration quality was considered acceptable for all major elements investigated in this study.

### Comparing calculated chemistry to bulk chemical measurements

All crushed materials were present in sufficient quantities to permit bulk chemical analysis by XRF, as the method of crushing after fragmentation permits the production of excess material from recovered clasts.

The compositional data are displayed in Supplementary Table 2. There is negligible compositional variation in the materials, with the average relative variation for each element across all four samples being less than $1 \pm 1\%$. Accordingly, it is clear that bulk compositional variation in the crushed samples cannot account for the trends observed in the XPS analysis (Supplementary Table 1).

There was insufficient material to permit XRF analysis of the fragmented materials by the same technique. We therefore performed micro-XRF analysis[52] on all crushed and fragmented materials. Although the technique has been successfully utilised to investigate lake sediments, we found that, relative to the bulk XRF measurements, Na was consistently 1.2 times higher and Mg was consistently 0.6 times lower in the micro-XRF dataset (Supplementary Table 2). In contrast, all other elements were in close agreement with the values measured by the bulk technique.

Considering the relative consistency in micro-XRF results between fragmented and crushed samples, we assume that the bulk chemical composition of the crushed samples offers a reasonably proxy for the fragmented materials. Accordingly, in our discussion and in Fig. 2a, we presume that all samples have the same average bulk chemical composition measured by standard XRF analysis and displayed in the (lower) right-hand column of Supplementary Table 2.

We further interrogate this assumption by reference to the QEMSCAN data. Our analysis shows some differences in phase abundance across the samples, principally in the abundance of plagioclase and matrix glass (Supplementary Table 3). The differences present the possibility that partitioning of different elements into different phases amplifies the significance of even small variations in mineral abundance and may account for the variations observed by XPS.

To determine the importance of this amplification, we calculate the bulk composition of all samples using the QEMSCAN phase abundance data (Supplementary Table 3) and EPMA compositional analysis for each of the observed mineral phases (Supplementary Table 4, Supplementary Table 2). Glass measurements with a 10-µm defocused beam were challenging due to the heterogenous glass microtexture and high microlite number density. Although some measurements of matrix glass composition and boundary-layer glass composition were obtained with a defocused beam, the pristine chemical composition of the glass is inhibited by the abundance of Fe-rich nanolites and/or nm-scale immiscible globules within the matrix glass together with the potential for subsurface mineral contributions and smearing effects. Accordingly, we used iterative goal-seeking in Microsoft Excel to determine whether there was a single glass composition which could, when considered as a component alongside the known mineral phases, account for the chemical compositional data measured by bulk XRF.

Convincingly, all variations in mineral phase abundance across the crushed samples measured by QEMSCAN yield a composition that, when combined with a single Fe-rich, Na-rich, Mg-free silicic glass phase, can account for the bulk chemical composition of the XRF data with minimal error (Supplementary Table 5). Notably, when compared to the data points obtained for the boundary-layer glass and with EPMA matrix glass measurements, the calculated composition is neither unreasonable nor incompatible with the measurements (Supplementary Table 5). When we incorporate the calculated composition into the same calculations for the fragmented samples in Supplementary Table 6, we see the same constancy of composition as observed for the crushed glasses.

Based on the above, we conclude that (i) we may consider the average bulk chemical composition for the crushed materials as representative of the fragmented materials and (ii) that the trends we observe by XPS cannot be explained by variations in the abundance of mineral and glass phases across the samples.

## Data availability

QEMSCAN data and analysis files generated in this study have been deposited in the Dryad data repository, accessible at https://doi.org/10.5061/dryad.5x69p8db2 All other datasets generated in the study are summarised in Supplementary Tables 1–5. Data analysis files are available by request to the corresponding author.

## Code availability

ImageJ macros used for QEMSCAN analysis are deposited in Github, at https://github.com/hornbya/QEMSCAN_ImageJ_macros.

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

## Acknowledgements
This work was supported by the ERC Advanced Grants EVOKES (247076) and EAVESDROP (834225). A.J.H. was supported by the European Union via a Marie Curie Individual Fellowship (AVAST 653900) and by a NASA IDS grant "Volcanic Ash and Its Impacts the Earth System" (19-IDS19-0083). J.E.K. was supported by an Early Career Fellowship of the Leverhulme Trust (ECF-2016-325). B.S. was supported by the Deutsche Forschungsgemeinschaft (DFG, German Research Foundation)—Project-ID 364653263—TRR 235. Richard Worden and FEI company are thanked for access to the QEMSCAN apparatus and Johan Rydberg for micro-XRF measurements. This work made use of the Cornell Centre for Materials Research facilities, which are supported through the NSF MRSEC programme (DMR-1719875). We thank Pierre Delmelle, Fabian Wadsworth and Yan Lavallée for discussions during the early phase of the study. Any use of trade, firm or product names is for descriptive purposes only and does not imply endorsement by the U.S. Government.

## Author contributions
A.J.H.: Conceptualisation, methodology, formal analysis, investigation, resources, data curation, writing (original draft, review and editing), visualisation, supervision, funding acquisition. P.M.A.: Conceptualisation, methodology, validation, formal analysis, investigation, resources, writing (original draft, review and editing), visualisation, supervision, funding acquisition. D.E.D.: Conceptualisation, methodology, investigation, writing (review and editing). S.D.: Investigation (XPS), methodology, writing (review and editing). J.E.: Sample collection, investigation (SEM-EDX and SE), writing (review and editing). J.E.K.: Investigation (QEMSCAN), writing (review and editing). C.C.: Investigation (SEM-EDX and BSE), writing (review and editing). U.K.: Sample collection, resources, writing (review and editing). B.S.: Investigation (shock tube experiments), writing (review and editing). J.E.P.U.: Investigation (QEMSCAN), writing (review and editing). D.B.D.: Writing (review and editing), project administration, funding acquisition.

## Competing interests
The authors declare no competing interests.
