## [Peer Review File · Nature Communications]

REVIEWER COMMENTS

Reviewer #1 (Remarks to the Author):

This manuscript is about the deflection of fractures in a multiphase andesite creating particles with different surface chemistry from the bulk. In the andesite sample studied, fracture propagation tends to focus in compositional boundary layers surrounding mafic crystals and tends to avoid (or thin within) areas with nanoscale Fe-rich blobs formed from silicate liquid immiscibility. The novel research is well-documented and results and conclusions are original and interesting. I enjoyed reading it and my only concern is that the key properties and textures of the one sample studied might not be common in the magmas fragmented in the eruptions for which we most need to understand the surface chemistry.

The abstract starts with the motivation and impact of the study: "Large explosive volcanic eruptions produce vast quantities of silicate ash that disperse through the atmosphere and deposit onto Earth's surface. During transit, ash surfaces are altered by... This altered surface likely mediates fundamental environmental interactions, including reactions with ozone, atmospheric ice nucleation, fertilisation of iron-limited ocean waters..." I was not convinced that the nanoscale textures key to the present study are present in a significant fraction of the pre-eruption magmas that fragment in large explosive eruptions. The sample studied is a 200 cc (low-vesicularity? from the images) andesite block from a pyroclastic flow deposit, which likely has a substantially different cooling (and undercooling) history from the magma that typically forms the ash in large explosive eruptions. I do not know, but am concerned that this could affect the potential for silicate liquid unmixing before fragmentation and that the sample in question may even have undergone this unmixing after fragmentation. Looking through (but not reading in detail) the cited papers I find that such unmixing in volcanic products has been in lava flows (mostly basalt) and lava lake samples, and I didn't see anything to give me any more confidence that would occur pre-fragmentation in large explosive eruptions. I would be pleased if the authors could provide their reasoning on the conditions for which they think this would occur. The compositional boundary layers around the mafic crystals can more easily be argued to be common, and perhaps these could have an important effect on fracture propagation and so surface chemistry even if melt immiscibility is not common in magmas forming large explosive eruptions. One could also look for these textures in ashes from large volcanic eruptions, but that would I think be beyond the scope of this study.

Overall, it is an excellent paper that could influence thinking in the field. But for a high-profile journal I would like more confidence in the link with the impact of the results.

Minor points:

line 190-192. The areas with Fe-rich droplets appear too broad to be compositional boundary layers - perhaps you could explain you could expand on your reasoning.

line 140 "show shows"

line 154 "through a particle" or "through particles"

line 162 "creates" rather than "creating"?

Reviewer #2 (Remarks to the Author):

The manuscript presents a detailed analysis of the micro- to nanoscale structures related to diffusion-limited microlite crystallization and their role in determining the surface composition of pyroclastic material. The authors claim that the deriving textural inhomogeneities can strongly control the fragmentation process and fracture distribution, resulting in a measurable disparity between surface and bulk composition of the fragments. To sustain their conclusions, the authors present a large dataset collected with several different analytical methodologies, covering bulk to surface analyses at different scales.

While the presence of compositional boundary layers around crystals is not novel and had already been discussed by other authors, the possible role of an uneven glass composition in controlling fracture formation is here presented convincingly (to my knowledge for the first time), although some doubts still remain about the real effect of this process on determining the observed differences between surface and bulk composition of the pyroclasts.

In the following, I raise several points that the authors should clearly address in order to justify their conclusions.

1- Analytical data derive from a variety of different methodologies. As commonly accepted, differences may be present between the results of the analyses for the different accuracies of the methods and the different problems intrinsic to each method. In particular, the presented data derive from bulk analyses of powdered samples (XRF, micro-XRF), surface analyses (XPS), analyses of polished samples (EPMA, QUEMSCAN, SEM-EDX). Two problems hence immediately arise before considering the comparison of these data:

a. Is accuracy of the different dataset really comparable, especially for those elements (like for ex. Mg or Na) which may present important problems for each different methodology?

b. Can the analyses of the external surface of the grains be compared with analyses of polished surfaces or with analyses of crushed, successively sieved and disproportionated (only the 63-90 μ m class was considered) and then powdered samples?

2- Another important methodological issue is related to the starting material used for the analysis. The author used a bomb from the PDC deposits of one of the largest recent events of Tungurahua. Bombs from this type of eruptions are generally characterized by important internal gradients in vesicularity and

groundmass crystallinity (see for ex. Gaunt et al., JVGR 2020), that make the use of this material for evidencing small differences poorly appropriate. A detailed description of the starting material, with a quantification of the main textural features, would be absolutely needed.

3- Interpretation as a result of Fe-rich silicate liquid immiscibility of the white spots in the glass is not fully convincing. Larger spots are in fact not spherical (Fig. 4a,b Extended Data) and resemble more oxide nanolites/microlites.

4- The calculated ideal bulk glass composition is enriched in Fe and Mg respect to EPMA data on the glass, that, on the contrary, are suspected to “likely include variable contribution of immiscible globules” (Extended data, Table 5). How can be this explained? Does the ideal glass comprise also the immiscible melt portion? This is very important, as it clearly influences also the calculated near surface composition (Extended data, Table 6).

5- While diffusion-limited crystallization, and related compositional boundary layers, are very clear for Px, diffusion-limited, disequilibrium growth for plagioclase is not clear. From images, Plag shows microlites with a regular prismatic habitus and no clear boundary layer, differently from Px that shows skeletal shapes. So, the glass showing the white spots could just be fully representative of the residual melt.

6- Authors evidence differences in crystal content between fragments derived from shock-tube experiments and mechanically fragmented material, with a slight enrichment in glass in the former. This does not surprise, as decompression-driven fragmentation mainly produce fine ash from rigid fragmentation of the septa separating bubbles or from the most vesicular portions of the starting material. Septa are in general depleted in crystals respect to the less vesicular, dense portions of the bombs. For this reason, a detailed analysis of the starting material (in terms of vesicularity, crystallinity and related inhomogeneities) should be presented and discussed.

7- Images from Fig. 2 and Fig. 4 Extended Data clearly show that many fractures cut across the “globule-rich” zones (Fig. 4a, f), while other follow the contact between the two different glasses (Fig. 2c). If these fractures represent original features, they clearly demonstrate that fragmentation should result in different types of external surfaces, characterized by large inhomogeneities (surfaces comprising both Fe-poor and Fe-rich glass portions) or by contrasting surfaces (surfaces of Fe-poor glass facing surfaces of globule-carrying glass), that do not explain the presence of a constant pattern of differences as evidenced by the results.

8- The choice of analyzing only fragments of a given dimension (63-90 μm), although justified by the need of establishing a common analytical procedure, could induce a bias in the generalized interpretation of the results. If the suggestion that glass composition and rheological properties are important factors in determining the geometry and spatial distribution of fractures, the relative scales of glass inhomogeneities (in terms of 3D volume) assume an important role in determining the surface features of fragments of different size. For this reason, the final suggestion that environmental effects of reactive glass surfaces is possibly different from that we can estimate basing on bulk composition, becomes weak. In this case I suspect that, as a general effect, consideration of the entire spectrum of particle sizes would probably suggest that surface composition is very close to the bulk composition.

9- The starting material possibly derives from fragmentation of a plug with a complex thermal history, which could justify the presence of disseminated Fe-rich spots in the glass similar to that observed in the re-heating experiments of d’Orlano et al. (2013) and Deardorff and Cashman (2017). The observed homogeneity of the starting material before and after heating at 850°C could derive from this fact and from the different duration of the heating experiments (not specified here).

10- The discussion is generally poorly focused. It is not clear if the intent of the authors is to focus onto the role of glass inhomogeneities in driving the fragmentation process or onto the differences of surface vs. bulk composition and their relationships with environmental effects of ash dispersal. In the two cases, a direct comparison of experimentally derived fragments with natural ash should also be presented, in order to verify the suggested conclusions. If the authors want to focus onto environmental effects and the importance of the observations at the nanoscale (“without nanoscale observation and consideration of the primary, unaltered ash surface, our speculations would be drastically different”), a deep study of the surface composition of particles of different sizes would be absolutely needed. In fact, if glass inhomogeneities exert an important control of fragmentation dynamics and geometry, the scale of these inhomogeneities could strongly control the surface composition of the fragments, with the fine-grained material being possibly different (and heterogeneous) respect to coarse-grained material. The overall ash-environment interactions could be in this case more dependent on bulk composition of the ash, even if still different at the grain scale.

11- The observed presence of immiscible globules in the Fe-enriched portions of the groundmass glass cannot be considered a general feature driving formation of pyroclastic material and controlling surface composition, but it is possibly mainly restricted to specific magma compositions (mafic to intermediate) showing a large microlite, disequilibrium-controlled crystallization. These conditions are not typical for example of the silicic magmas that mainly drive large eruptions with a strong environmental impact, and whose products are generally characterized by microlite-free, compositionally homogeneous glass.

Reply to reviewers

Reviewer #1:

This manuscript is about the deflection of fractures in a multiphase andesite creating particles with different surface chemistry from the bulk. In the andesite sample studied, fracture propagation tends to focus in compositional boundary layers surrounding mafic crystals and tends to avoid (or thin within) areas with nanoscale Fe-rich blobs formed from silicate liquid immiscibility. The novel research is well-documented and results and conclusions are original and interesting. I enjoyed reading it and my only concern is that the key properties and textures of the one sample studied might not be common in the magmas fragmented in the eruptions for which we most need to understand the surface chemistry.

We thank the reviewer for their positive feedback. Our intention with this submission is to demonstrate that magma fragmentation and the resulting surface chemistry of volcanic ash particles are affected by micro-to-nano-scale chemical and textural inhomogeneities in the matrix. These features vary widely from one eruption to the next, but similar textures produced by rapid groundmass crystallization in pre-eruptive magmas are widely reported in the literature, all of which are envisaged to have similar implications for ash surfaces. Removal of the focused immiscibility discussion now allows this point to shine more clearly.

The abstract starts with the motivation and impact of the study: “Large explosive volcanic eruptions produce vast quantities of silicate ash that disperse through the atmosphere and deposit onto Earth’s surface. During transit, ash surfaces are altered by... This altered surface likely mediates fundamental environmental interactions, including reactions with ozone, atmospheric ice nucleation, fertilisation of iron-limited ocean waters...” I was not convinced that the nanoscale textures key to the present study are present in a significant fraction of the pre-eruption magmas that fragment in large explosive eruptions.

This is a good point, and we have made clear that the environmental implications of our study apply to a set of compositions and eruption styles for which at least partial pre-eruptive matrix crystallization is expected. These include a wide range of arc eruptions where shallow storage of pre-eruptive magma is common, and where production of large quantities of volcanic ash are regularly produced via explosive eruptions and PDCs (such as at Tungurahua). Although not as concerning from a global perspective as eruptions fed directly from a deeper magma chamber, these types of eruptions have long-lived acute regional impacts and represent the largest global source of volcanic ash on decadal to millennial timescales.

Also, we reiterate that the exact textures observed here may not be reproduced in other magmas or eruptions, however disequilibrium textures within volcanic melts are very common in volcanic ash deposits due to the highly disequilibrium conditions leading to eruption. Therefore, the implications of textural heterogeneities in pre-eruptive magma on magmatic fragmentation and subsequent surface chemistry that we document here is likely to be widely applicable despite the diversity of magmatic micro-to-nanotextures and speaks to our conclusion that the bulk ash chemistry may not accurately represent volcanic ash surfaces. For example, the two other reviewers brought up an additional mechanism that may generate lack of parity between surface and bulk chemistry in bubble-rich magmas.

We address the legitimate concern raised here via

1 – New high-magnification imagery and image analysis of natural airfall ash samples from the Tungurahua 2006 eruption (Figure 3, Supplementary figure 5-6) together with detailed discussion of the eruptive products and eruption sequence for the Tungurahua eruption in the Methods. We show that 15-20% of the natural volcanic ash particles contain the same set of textures as the experimental pyroclasts, and we contextualize this as resulting from mixed airfall deposits from co-PDC ash and ash from the magmatic plume.

2 – Clarified statements on the applicability of our results to eruptions in which groundmass crystallization plays an important role, for example in diverse arc volcanism, dome eruptions and accompanying hazards such as PDCs. We have also removed “Large” as the first word of the Abstract to modify perception from the outset.

The sample studied is a 200 cc (low-vesicularity? from the images) andesite block from a pyroclastic flow deposit, which likely has a substantially different cooling (and undercooling) history from the magma that typically forms the ash in large explosive eruptions. I do not know, but am concerned that this could affect the potential for silicate liquid unmixing before fragmentation and that the sample in question may even have undergone this unmixing after fragmentation. Looking through (but not reading in detail) the cited papers I find that such unmixing in volcanic products has been in lava flows (mostly basalt) and lava lake samples, and I didn't see anything to give me any more confidence that would occur pre-fragmentation in large explosive eruptions. I would be pleased if the authors could provide their reasoning on the conditions for which they think this would occur.

The compositional boundary layers around the mafic crystals can more easily be argued to be common, and perhaps these could have an important effect on fracture propagation and so surface chemistry even if melt immiscibility is not common in magmas forming large explosive eruptions. One could also look for these textures in ashes from large volcanic eruptions, but that would I think be beyond the scope of this study.

We thank the reviewer for this discussion – indeed, the material studied in the sample is not characteristic of the ash producing the sub-Plinian eruption column in the paroxysmal phase of the eruption, but is inferred to represent the more crystal rich material involved in fire fountaining and subsequent collapse in PDCs. To address their concern regarding timings of unmixing, we have made two critical changes to the manuscript:

First, our textural analysis of natural samples within the main axis of the plume dispersion shows that 15-20% of the ash bore textures indistinguishable from our experimental block. This implies that any post-eruptive process forming the matrix textures we observe would likely have been common in the material fragmented in PDC and not unique to the block we sampled. Although some degree of textural variation during cooling is inevitable, given the wide variety of cooling rates involved during fire-fountaining activity, such ubiquity of texture argues against post-fragmentation development of the magma textures we concentrate on here. We have included discussion around this point in our third-to-last paragraph and the Methods.

Second, following reviewer feedback on the textures we previously attributed solely to silicate liquid immiscibility, we have permitted the interpretation of the bright nanoscale features as either/both nanolites or immiscible droplets. Exact classification is not possible for most of our imagery and is not essential in demonstrating that the nanoscale textural features alter the fragmentation process and ash surface chemistry. Therefore, we have

re-labeled the bright nanoscale phase as ‘nanolites and/or immiscible droplets’. This does not affect any of our conclusions or interpretation of the effects on fragmentation pathways, which already referenced the corresponding effects of nanolite crystallization and phase separation in glasses.

We agree with the reviewer in that further textural analysis of ashes is beyond the scope of this study. However, as part of the revision process we have compiled a list of case-examples that will form the basis of future work, and we thank the reviewer for challenging us on this point.

Reviewer #2:

The manuscript presents a detailed analysis of the micro- to nanoscale structures related to diffusion-limited microlite crystallization and their role in determining the surface composition of pyroclastic material. The authors claim that the deriving textural inhomogeneities can strongly control the fragmentation process and fracture distribution, resulting in a measurable disparity between surface and bulk composition of the fragments. To sustain their conclusions, the authors present a large dataset collected with several different analytical methodologies, covering bulk to surface analyses at different scales. While the presence of compositional boundary layers around crystals is not novel and had already been discussed by other authors, the possible role of an uneven glass composition in controlling fracture formation is here presented convincingly (to my knowledge for the first time), although some doubts still remain about the real effect of this process on determining the observed differences between surface and bulk composition of the pyroclasts. In the following, I raise several points that the authors should clearly address in order to justify their conclusions.

We appreciate the reviewer recognizing the novelty of the findings and are pleased that they found our discussion of process convincing.

1- Analytical data derive from a variety of different methodologies. As commonly accepted, differences may be present between the results of the analyses for the different accuracies of the methods and the different problems intrinsic to each method. In particular, the presented data derive from bulk analyses of powdered samples (XRF, micro-XRF), surface analyses (XPS), analyses of polished samples (EPMA, QUEMSCAN, SEM-EDX). Two problems hence immediately arise before considering the comparison of these data:

- a. Is accuracy of the different dataset really comparable, especially for those elements (like for ex. Mg or Na) which may present important problems for each different methodology?
- b. Can the analyses of the external surface of the grains be compared with analyses of polished surfaces or with analyses of crushed, successively sieved and disproportionated (only the 63-90 μ m class was considered) and then powdered samples?

We agree with the reviewer that these are valid concerns. It is for this very reason that we intentionally make our comparisons through multiple datasets and follow multiple lines of evidence to substantiate our explanations of the observed trends.

Regarding (a), in short, no, we do not claim accuracies are comparable across methods. This is a well-known issue, and much work is done to standardize methods and benchmark data in labs globally. As a general point, we have used methods that have been vetted and refined

over decades across many disciplines. The one technique we employed that is not well-established is micro-XRF, and we have opted not to use this data in any multi-method comparative calculations but to replace it with standard XRF measurements (described in the “Comparing calculated chemistry to bulk chemical measurements” section of the Methods).

Each technique has different sensitivities, and we attempt to follow best practice and acknowledge known issues and limitations in our data generation. For example, it is well known that Na mobility may be an issue for EPMA and SEM-EDX analyses, particularly of glasses. For this reason that we have not used a focused beam in EPMA measurements of matrix glass, and instead infer the composition using measurements which are well-established and often the best available for the purpose – for example, QEMSCAN is the highest resolution method available to simultaneously determine phase fractions and phase distribution over the scale required, and XPS allows measurements with the highest degree of specificity at the nm-scale ash surface. We were also careful to propagate the error associated with each dataset through consecutive sets of calculations and show the error bars in Figure 2.

Regarding the consequences of variable measurement accuracies, we take reassurance from the data and results themselves, where we rely heavily on trends and are careful not to discuss absolute changes. If, for example, the observed differences were wholly attributable to method dependencies, then we would not observe the consistent treatment-dependent differences that we see in Mg or Na, for example (see also Supplementary Discussion 1. The magnitude of those differences becomes debatable, but not the existence of differences themselves. That our results are congruous across multiple datasets leads us to conclude that the effect of absolute differences in accuracy is inconsequential to our broader conclusions.

Regarding (b), an initial point to make here is that XPS and QEMSCAN measurements of the external surfaces of grains were all made on the same 63-90 μm size fraction as all other measurements (e.g., XRF and SEM), following any fragmentation, crushing and sieving steps. To be clear, XRF and micro-XRF measurements were made on the sieved samples, but they were milled and prepared into pellets by the host labs prior to bulk chemistry measurements.

In Figure 1c, we measured phase fractions on single-pixel-wide outlines (e.g., max. 2-3 microns deep) of polished particle surfaces via QEMSCAN and compared these to XPS measurements of particle surfaces via average phase chemistry measured by EPMA and mass balance. Although the sensitivity of the QEMSCAN-EPMA based technique is undoubtedly lower than the XPS data, when we use a single calculated glass composition we see no major variations across eight different samples when compared to XPS vs XRF data (and where XRF is the discipline standard for measuring bulk chemistry). This can be seen by comparing Figure 1a and Figure 1c, where both absolute values and the relative variations between samples are well reproduced.

To reinforce this point, we include below and in Supplementary Figure 1 the data for XPS vs the bulk chemistry calculated from QEMSCAN-EPMA data, using the same key as Figure 1, which can be directly compared to the XPS vs XRF data.

2- Another important methodological issue is related to the starting material used for the analysis. The author used a bomb from the PDC deposits of one of the largest recent events of Tungurahua. Bombs from this type of eruptions are generally characterized by important internal gradients in vesicularity and groundmass crystallinity (see for ex. Gaunt et al., JVGR 2020), that make the use of this material for evidencing small differences poorly appropriate. A detailed description of the starting material, with a quantification of the main textural features, would be absolutely needed.

We thank the reviewer for making this important point and have now included a detailed set of images describing the texture and microtexture of the starting material in a new Figure 1. The block is reasonably isotropic, with large and slightly elongate pores presenting the greatest textural variations. Nevertheless, there is no consistent gradient in texture across different core locations or along core axes.

Importantly, the chemical variations that we highlight here are not primarily found between the eight samples we studied, but for different depth resolutions within the same sample. We created pyroclasts from cored samples over a range of pressures, temperatures and fragmentation method and measured consistent relative variations for the same elements at pyroclast surfaces in every case, suggesting that core samples were sufficiently similar to validate our results.

3- Interpretation as a result of Fe-rich silicate liquid immiscibility of the white spots in the glass is not fully convincing. Larger spots are in fact not spherical (Fig. 4a,b Extended Data) and resemble more oxide nanolites/microlites.

Together with comments from other reviewers, we have chosen to revise our description of the bright nanoscale phase as ‘nanolites and/or immiscible droplets.’ We have included new images of natural and experimental material that show these features. In some cases, a rounded or hemispherical shape is apparent but in other cases they appear more like

nanolites. It is important to point out that the exact classification is not important to the conclusions of our study – that nanoscale disequilibrium features in the matrix affect the surface chemistry. Nanolites and immiscible droplets have similar impacts on fracture toughness crack propagation, therefore our interpretation remains unchanged and the effect of nanoscale disequilibrium textures can be more broadly implied.

4- The calculated ideal bulk glass composition is enriched in Fe and Mg respect to EPMA data on the glass, that, on the contrary, are suspected to “likely include variable contribution of immiscible globules” (Extended data, Table 5). How can be this explained? This is very important, as it clearly influences also the calculated near surface composition (Extended data, Table 6).

Yes, the ideal glass models the glass composition including all phases that are below the resolution of the QEMSCAN analysis. Higher Fe/Mg may be expected in the ideal glass, since EPMA spots were chosen to attempt to avoid other phases/globules as far as possible. However, we note that the Mg concentration calculated in the ideal glass is significantly lower than the in the measured glass – 0.22 wt% vs 1.16 wt%. As we note in the manuscript, this strengthens our findings of extreme Mg depletion: “Indeed, notwithstanding that use of the calculated Mg concentration (0.22 atm.%) instead of defocused EPMA measurements (0.6–1.16 atm.% Mg) of the matrix glass reduces the discrepancy between microscale and nanoscale surface compositions, we find nanoscale surface depletions of Mg by a factor of 0.5–0.05 across all samples (Figure 2c).” We thank the reviewer for bringing up this point and have clarified that the ideal glass chemistry theoretically includes sub-resolution phases that are measured by XRF but not by QEMSCAN in the manuscript and the Table caption.

5- While diffusion-limited crystallization, and related compositional boundary layers, are very clear for Px, diffusion-limited, disequilibrium growth for plagioclase is not clear. From images, Plag shows microlites with a regular prismatic habitus and no clear boundary layer, differently from Px that shows skeletal shapes. So, the glass showing the white spots could just be fully representative of the residual melt.

Although plagioclase crystals may not have grown at the same rate as pyroxenes and therefore development of boundary layers of comparable size is not necessary, we suggest that an Fe-enriched boundary layer may have existed around plagioclase crystals as commonly observed in mafic products. However, a boundary layer expected to be enriched in Fe and Mg may ‘disappear’ upon nanolite or immiscible globule formation, with the enriched concentrations of these elements partitioning into the nanophases. The former presence of an Fe-enriched halo is suggested by the common observations of coarser bright, nanoscale features directly surrounding plagioclase crystals. The inference here is that the compositional boundary layer predated nanolite crystallization or unmixing of the melt, as inferred by Honour et al. (2019). The other possibility is that nanolites or immiscible globules close to plagioclase boundaries grew or amalgamated at a faster rate than more remote matrix glass due to higher Fe concentrations in a boundary layer. We thank the reviewer for the observation but it does not directly affect any part of the manuscript discussion or conclusions and no changes have been made.

6- Authors evidence differences in crystal content between fragments derived from shock-tube experiments and mechanically fragmented material, with a slight enrichment in glass in the former. This does not surprise, as decompression-driven fragmentation mainly produce

fine ash from rigid fragmentation of the septa separating bubbles or from the most vesicular portions of the starting material. Septa are in general depleted in crystals respect to the less vesicular, dense portions of the bombs. For this reason, a detailed analysis of the starting material (in terms of vesicularity, crystallinity and related inhomogeneities) should be presented and discussed.

As mentioned, we have now included a set of images showing the macrotexture of the starting material. Bubbles have a wide size distribution, but are volumetrically dominated by large, rounded bubbles. This is a very interesting point made by the reviewer, but the measurement of interest for this study is the surface normalized to the bulk. We find consistent differences in this metric across all samples, suggesting that the fragmentation mode exerts an inferior, yet observable, influence on surface chemistry. A more detailed examination of the effects of fragmentation mode and temperature and pressure conditions would make for an interesting follow-up study, but the data we present here suggests that these effects are significantly milder than the surface-bulk effect that we document here – we elaborate on this in Supplementary Discussion 1.

7- Images from Fig. 2 and Fig. 4 Extended Data clearly show that many fractures cut across the “globule-rich” zones (Fig. 4a, f), while other follow the contact between the two different glasses (Fig. 2c). If these fractures represent original features, they clearly demonstrate that fragmentation should result in different types of external surfaces, characterized by large inhomogeneities (surfaces comprising both Fe-poor and Fe-rich glass portions) or by contrasting surfaces (surfaces of Fe-poor glass facing surfaces of globule-carrying glass), that do not explain the presence of a constant pattern of differences as evidenced by the results.

There will certainly be a range of external surfaces. The consistent patterns in the data show that there is a preferred mode, however, and our argument is that the best explanation for the clear pattern of differences that we observe is

- i) via the fractures preferentially passing through the compositional boundary layer close to, but not directly along the pyroxene crystal surface (and close to, but not directly at the interface between globule-free boundary layer glass and globule-bearing matrix glass). XPS measurements at nanoscale resolution would produce an Fe-depleted signal for both surfaces in this case, whereas EDX measurements would produce the kinds of signals described in the comment.
- ii) via fractures deviating around the Fe-rich nanoscale phase and terminating in globule-rich glass, thereby ‘insulating’ these high Fe (and Mg) zones from nm-scale particle surfaces.

These two mechanisms are highlighted in the manuscript. The textural evidence that we present alongside our discussion and referencing of fracture propagation supports these mechanisms as plausible, though we welcome robust discussion of alternative explanations.

8- The choice of analyzing only fragments of a given dimension (63-90 μm), although justified by the need of establishing a common analytical procedure, could induce a bias in the generalized interpretation of the results. If the suggestion that glass composition and rheological properties are important factors in determining the geometry and spatial distribution of fractures, the relative scales of glass inhomogeneities (in terms of 3D volume) assume an important role in determining the surface features of fragments of different size. For this reason, the final suggestion that environmental effects of reactive glass surfaces is possibly different from that we can estimate basing on bulk composition, becomes weak. In

this case I suspect that, as a general effect, consideration of the entire spectrum of particle sizes would probably suggest that surface composition is very close to the bulk composition.

Indeed, for particles smaller than the textural inhomogeneities, we would not expect any of the biases we document here. In this case, the features that we describe are on the order of 10s of nm up to several microns. In explosive volcanic eruptions, the majority of the mass of pyroclasts is in coarser size fractions, where the effect of textural inhomogeneities on fragmentation may be expected to become apparent. It is important to recognize that the composition of ultrafine ash particles is not well established – at all – and may be significantly different to the bulk for different reasons. In addition, different sized particles are spatially separated due to different settling velocities and therefore size-dependent variations in surface chemistry would likely impact different receiving environments. The broader implication that finer or coarser fractions would ‘balance’ any environmental impacts from disparity in surface-to-bulk composition appears to be purely speculative and further targeted work would be required to test the hypothesis, which is of interest given the findings of this study but beyond scope at present.

9- The starting material possibly derives from fragmentation of a plug with a complex thermal history, which could justify the presence of disseminated Fe-rich spots in the glass similar to that observed in the re-heating experiments of d’Oriano et al. (2013) and Deardorff and Cashman (2017). The observed homogeneity of the starting material before and after heating at 850°C could derive from this fact and from the different duration of the heating experiments (not specified here).

We thank the reviewer for bringing up this important point. During these revisions we have examined two natural ash samples collected at 2 km and 12 km from the vent at Tungurahua. We have now presented images in Figure 2c and Supplementary Figure 2 that show texturally indistinguishable features in 15-20 % of the natural ash from the Tungurahua 2006 eruption. Detailed analysis of the eruption sequence and products (now detailed in the Methods) shows that the textures in our experimental block are likely shared by the products of fire-fountaining during the pre-paroxysmal phase of the eruption on August 16th 2006. Ramparts built by this activity repeatedly collapsed forming PDCs, and the airfall ash during the final paroxysmal phase represents a mixture of ash from the magmatic plume and co-PDC ash (see e.g. Eychenne et al., 2012]. The latter ash is inferred to bear the crystal-rich matrix glass found in our experimental block, while the magmatic eruption plume contained ash bearing crystal-poor and Fe-rich matrix glass. These contrasting textures are consistent with a magma that stalled at shallow depth, which was mobilized by an injection of fresher magma having a microlite-poor and Fe-enriched matrix glass. These processes of relatively mafic magma injection and multi-level magma storage are near-ubiquitous in arc volcanoes, as are the development of disequilibrium melt textures prior to eruption, therefore we argue that the fracture-deflecting processes that we discuss may be common in eruptions of arc volcanoes, especially during the opening phases.

10- The discussion is generally poorly focused. It is not clear if the intent of the authors is to focus onto the role of glass inhomogeneities in driving the fragmentation process or onto the differences of surface vs. bulk composition and their relationships with environmental effects of ash dispersal. In the two cases, a direct comparison of experimentally derived fragments with natural ash should also be presented, in order to verify the suggested conclusions. If the authors want to focus onto environmental effects and the importance of the observations at

the nanoscale (“without nanoscale observation and consideration of the primary, unaltered ash surface, our speculations would be drastically different”), a deep study of the surface composition of particles of different sizes would be absolutely needed. In fact, if glass inhomogeneities exert an important control of fragmentation dynamics and geometry, the scale of these inhomogeneities could strongly control the surface composition of the fragments, with the fine-grained material being possibly different (and heterogeneous) respect to coarse-grained material. The overall ash-environment interactions could be in this case more dependent on bulk composition of the ash, even if still different at the grain scale.

We agree with the reviewer that the discussion had lost a little focus and became somewhat dominated by the mechanics of texture formation and the fragmentation process. We have condensed some of this description and moved discussion on the rheological effect of unmixing into a new Supplementary Discussion 3. This serves to help highlight the environmental implications and the core message of the work – that the surface composition does not match the bulk and microscale.

The second point is addressed by the new images of natural ash, where the same microtextures are documented.

The final point, this was made earlier by the reviewer (Point #8). Although it is possible that the speculation of the reviewer is accurate, this has not been established, and the mass of particles finer than the inhomogeneities is far smaller than that of the coarser. That being said, this is perhaps a limiter on the application to long-lived and long-traveled ash clouds, where ash grain sizes distribution approaches the size of the nanotextures.

In principle, the apparent position of the reviewer gathered through Points 8 and 10 perfectly outlines the need for studies like this one, and the potential for follow-up work that this first study generates. We contend that we have shown that surface chemistry is inhomogeneous and submitted a strong viable unit of observation-explanation-implications for particles across a size range that have significant environmental relevance for most eruptions. There is no reason to think the mechanisms we discuss in the manuscript should not apply to particles coarser than the texture size. On the other hand, future work investigating ultrafine particles will be very interesting and relevant, especially as they have been hitherto neglected by the volcanology and atmospheric science communities.

11- The observed presence of immiscible globules in the Fe-enriched portions of the groundmass glass cannot be considered a general feature driving formation of pyroclastic material and controlling surface composition, but it is possibly mainly restricted to specific magma compositions (mafic to intermediate) showing a large microlite, disequilibrium-controlled crystallization. These conditions are not typical for example of the silicic magmas that mainly drive large eruptions with a strong environmental impact, and whose products are generally characterized by microlite-free, compositionally homogeneous glass.

We agree in large part to this comment and have replied to a similar critique from Reviewer 1. For large silicic eruptions the mechanisms that we investigate here are not likely to be relevant. However, there are a broad range of eruption types, magnitudes and styles where these textures are likely to modify fracture paths and, subsequently, pyroclast surface chemistry and environmental impacts. These include dome eruptions which may produce

PDCs, mafic to intermediate arc eruptions, plume-fed explosive eruptions and mafic Plinian eruptions. While it is tempting to presume that impacts scale with absolute eruption size, impacts are inherently variable; eruptions affect environments and populations over a range of scales and durations and are also controlled by factors independent of an eruption, including ecosystem recovery. Given the recurrence times of the largest eruptions, in many cases it may be more relevant to understand the impacts of smaller, more frequent eruptions. As in our response to Reviewer 1, eruptions at mafic-to-intermediate arc volcanoes would appear to be most relevant in this case.

Dr. Heather Wright's USGS internal review comments:

Review of **Nanoscale silicate melt textures determine volcanic ash surface chemistry** by Hornby and others.

This paper discusses textural and chemical signatures of volcanic ash surface chemistry. The methodology is appropriate, but conclusions are not sufficiently novel for a Nature article or are insufficiently supported by data. The assertion that surface chemistry differs from bulk composition is unsurprising, given assumed fragmentation mechanisms, the multiphase character of volcanic pyroclasts, and the zoning common within phenocryst phases (see below). The assertion that Fe-oxide liquid immiscibility is more common than previously thought and controls fragmentation is not sufficiently well documented here.

We appreciate the reviewer's comments, and the inferred likelihood of surface-to-bulk chemical disparity, with which we agree. However, here we show that the disparity increases closer to the pyroclast surface and that chemical and physical heterogeneity in the magmatic melt can cause such surface chemistry variations. In addition, although intuitively likely, surface-to-bulk discrepancies have been neglected by much of the literature to date, and in the few cases where they have been demonstrated, no mechanisms or constraints on the scale of the phenomenon have been studied.

We have reconsidered our attribution of silicate liquid immiscibility for all the bright nanoscale features we document, and instead describe them as 'nanolites and/or immiscible droplets.' In either case, the effects on fracture toughness and fracture propagation are unchanged, and this highlights that a range of disequilibrium texture are likely to contribute to the surface chemistry of volcanic ash particles, which is the essential conclusion of our study.

I'm surprised to hear that studies have assumed that ash particles have surface chemistry similar in composition to bulk analyses. Fragmentation of bubbly melt (cf. review of Gonnermann et al. 2015), for which bubble wall thinning causes rupture or decompression induces fracture, should produce ash grains with surface chemistry similar in composition to groundmass composition, not bulk composition. As you state in the caption for Supplementary Figure 4, fractures redirect around crystals in many cases.

Although we believe that pyroclast surfaces are likely to differ from the bulk in many cases, this has not been investigated or published in the literature and the mechanisms by which this can occur have not been studied. We thank the reviewer for highlighting this additional mechanism. We believe the pervasive disequilibrium textures that we investigate here will differ with composition, eruptive style, and magma ascent and storage history. However, where they are present, the effect may well be similar – to alter the surface chemistry relative to the bulk. The bulk-surface equivalence assumption has been predominant in the ash-gas reaction and environmental interaction literature and is a simplification that we are explicitly investigating here; we find a significant variation likely due to micro-to-nano textural variations in the pre-eruptive magmatic melt.

Fractures occurring at the interface between melt and crystal, effectively liberating the crystal, is relatively commonly appreciated, but the redirection of fractures very close to crystals in the boundary-layer glass, as described here, is not. This gives rise to a new appreciation of highly depth-sensitive chemical disparity that we document.

I'm not convinced, however, by implied differences between nm-scale surface chemistry and um-scale surface chemistry. The calculations of um-scale surface chemistry presented here rely on multiplication of QEMSCAN phase percentages by an average composition of each phase (ideal glass) and subtraction from XRF totals, correct?

Correct – this is the calculation of um-scale surface chemical variation from the bulk, which we assume is what is intended.

How does crystal zonation affect calculations that use average composition to represent each phase (where zoned crystals usually have higher Mg and Ca than cores)? For example, your supplement includes plagioclase phenocryst analyses with CaO contents from 9 to 14 wt%. I'd expect crystal cores to be underrepresented in grain surfaces.

This is a valid point, and our calculated micron-scale surface composition may be somewhat skewed toward the bulk as a result. The magnitude of this effect is difficult to assess, but this logic implies that the inferred 'overrepresentation' of zoned crystal rims via crystal boundary fracture would increase the difference in Mg concentration between the micron-scale to nm-scale composition than previously supposed. However, the similarity between the micron-scale surface (QEMSCAN) to nanoscale comparison (Figure 2c) and the bulk micron-scale (QEMSCAN) to nanoscale comparison (new Supplementary Figure 1) suggests this has a minor overall effect.

Further, the authors state that "The calculated matrix composition is compatible with glass measurements obtained with a defocused beam". But the two differ significantly for several elements, by the largest amounts for Mg, Fe, and Ca (Supplementary table 5: matrix glass vs. ideal glass).

As a result, it's very difficult to estimate the consequent uncertainty in um-scale chemistry. Given the lack of uncertainty for um-scale chemistry, it is impossible to determine whether the conclusions about difference between nm-scale average surface chemistry significantly differ from um-scale chemistry.

Differences between the calculated and measured matrix glass compositions are expected, since the defocused beam measurements included varying, and unavoidable contributions from microlites, and boundary layer glass. The differences mentioned here in Mg, Fe and Ca are now explicitly invoked in the manuscript. In the same section, we describe that use of the calculated matrix composition ('ideal glass') as opposed to EPMA measurements substantially reduces the discrepancy for Mg, the element with the greatest variation. Therefore, any incorporation of measured matrix chemistry would increase the chemical variation between um-scale and nm-scale glass. For this reason, we state that the nm-scale composition cannot be directly linked to phase fractions present at the um-scale surface.

The relevant section of the manuscript now reads as follows:

“Although the calculated matrix composition is broadly compatible with glass measurements obtained by EPMA with a defocused beam (Supplementary Table 5, Supplementary Table 3), variations in Ca, Fe and Mg are noted, likely due the varying contribution of sub-resolution features in EPMA measurements. However, with these data and calculated values, we show that the microscale mineral variation in the near-surface region cannot account for variability in the nanoscale surface chemistry observed by XPS (Figure 1c, Supplementary Table 6, Supplementary Figure 1). Indeed, notwithstanding that use of the calculated Mg concentration (0.22 atm.%) instead of defocused EPMA measurements (0.6–1.16 atm.% Mg) of the matrix glass reduces the discrepancy between microscale and nanoscale surface compositions, we find nanoscale surface depletions of Mg by a factor of 0.5–0.05 across all samples (Figure 1c).

More information about the spatial scale of XPS measurements is warranted. In the Methods section, analytical area is 300x700 um, but I believe that the depth of analysis is just a few nm. Is that correct?

Yes - the 2-10 nm depth resolution of XPS measurements is now included in the manuscript in the methods as well as the main text.

You state that “Silicate liquid immiscibility and dynamic crystallisation are both understudied processes in magma textural evolution and published evidence for unmixing in calc-alkaline arc andesites like Tungurahua are uncommon”, but textures very closely resembling those in your Figure 2 are documented in previous studies of Tungurahua. See Wright et al. 2012, Figure 2E. As you say, they are found in Tarawera basaltic clasts (Sable et al. 2007). And Etna clasts (Szramek 2016). They are also documented in intermediate rocks, e.g., at Cotopaxi (Gaunt et al. 2017) and at NW Rota (Deardorff and Cashman 2017). But they are also characteristic of reheating textures, e.g., those documented in D’Oriano et al. 2012 and in Deardorff and Cashman 2017. See D’Oriano Figures 5f, 5o, 6k, 7f, 8d and 8f. Reheating/oxidation textures would be expected if any material is recycled at the vent during an eruption.

We agree that the textures documented for Sable et al. 2009 (Tarawera), Deardorff and Cashman 2017 (NW Rota and experimental reheating) and D’Oriano et al. 2013 look very similar to those documented here. We appreciate these references, especially as such textures are very rarely documented. Although we do not doubt that similar textures were imaged for the Tungurahua ash in Wright et al. 2012, the images presented do not clearly show the textures, possibly due to magnification/resolution issues. The same lack of clarity applies to textures documented in Gaunt et al. 2016 and Szramek et al. 2016.

In summary, we have found only a handful of convincing references to silicate liquid immiscibility in explosive products; in most cases, textures are unlabelled or apparently mislabelled and not included in results or discussion in the studies and are not effective documentation. Those studies that have documented and investigated immiscibility in depth have focused on ferrobasalts, tholeiitic series rocks and lavas or intrusive complexes. Due to the disequilibrium conditions preceding and accompanying volcanic eruption, dynamic crystallisation is likely to be the rule rather than the exception, however little account has

been given in the literature, with a few notable exceptions, such as the work of Bruce Watson [Watson, 1996; Watson & Müller, 2009] and Mollo and Hammer [Mollo & Hammer, 2017]

Therefore, we stand by our assertion that silicate liquid immiscibility and dynamic crystallisation are understudied processes in magma textural evolution and published evidence for unmixing in magmas outside of ferrobaltic and tholeiitic magma suites is uncommon. However, in response to these comments and those of other reviewers, we have revised our attribution of such textures from silicate liquid immiscibility only to ‘nanolites and/or immiscible droplets’.

As in our responses to Reviewers 1 and 2, the observation of these textures in 15-20% of the natural airfall ash speak against their origin from recycling or reheating and oxidation. We infer that these textures were commonly present in the materials produced by fire fountaining that repeatedly collapsed to form PDCs.

I agree with the authors’ statement that “silicate liquid immiscibility is more common in systems where dynamic crystallisation occurs, such as when magma ascent stalls at a shallow level prior to eruption or is subject to rapid temperature changes”. However, the connection to the authors’ next statement is not sufficiently supported in this manuscript. What is the support for the assertion that “silicate liquid immiscibility triggered by dynamic crystallisation may be more common than currently considered across a wide range of volcanic settings, potentially affecting the timing and intensity of fragmentation processes and playing an important role in determining the ash surface chemistry”? These textures are already documented in intermediate compositions and are associated with surface processes (reheating) too.

This statement came from the above-discussed scant published evidence for silicate liquid immiscibility, whereby textures are often unattributed or unclear. The spatial scale of immiscible textures is often at the nanoscale, and it is currently not common, even in textural studies, to specifically examine nm-scale features. Therefore, based on our literature search and the work conducted here, we stand by our statement that silicate liquid immiscibility is likely to occur more often and across a wider range of compositions than currently appreciated by many in the field, and we are working on such a survey. However, given that we have revised the description of the nanoscale features to include nanolites, we have removed much of the discussion around unmixing from the manuscript, including this assertion.

References.

- Deardorff, N., & Cashman, K. (2017). Rapid crystallization during recycling of basaltic andesite tephra: timescales determined by reheating experiments. *Scientific reports*, 7, 46364.
- D’Oriano, C., Pompilio, M., Bertagnini, A. *et al.* Effects of experimental reheating of natural basaltic ash at different temperatures and redox conditions. *Contrib Mineral Petrol* **165**, 863–883 (2013). <https://doi.org/10.1007/s00410-012-0839-0>
- Sable, J. E., HOUGHTONI, B., Wilson, C. J. N., & Carey, R. J. (2009). Eruption mechanisms during the climax of the Tarawera 1886 basaltic Plinian eruption inferred from microtextural. *Studies in volcanology: the legacy of George Walker*, (2), 129.

Szramek, L. A. (2016). Mafic Plinian eruptions: Is fast ascent required?. *Journal of Geophysical Research: Solid Earth*, 121(10), 7119-7136.

Wright, H.; Cashman, K.; Mothes, P.; Hall, M.; Ruiz, A.; Le Pennec, J.-L. 2012. Estimating rates of decompression from textures of erupted ash particles produced by 1999-2006 eruptions of Tungurahua volcano, Ecuador. *Geology* 40: 619-622.

Line by line comments

Line 56

How do disequilibrium textures promote violent explosions? Isn't it actually the rapid rheologic changes that promote this?

Rapid rheological changes and disequilibrium textural development are not necessarily independent – see for example the effect of nanolite crystallization on melt viscosity [Di Genova, Brooker, et al., 2020; Di Genova, Zandona, et al., 2020] and bubble nucleation [Cáceres et al., 2020], two of the most sensitive properties for explosivity. This sentence currently reads “.. causing rheological changes that often promote violent explosive activity”, which aligns with the Reviewer's statement.

Line 398 (Figure 3) Although phase percentages are similar (Figure 1b), compositions do not appear to be similar (Figure 1a).

Phase percentages are for the um-scale (Figure 1b) while XPS/XRF measurements compare the nanoscale (Figure 1a); therefore, this demonstrates the disparity between micro- and nanoscale chemistry relative to the bulk.

Line 425: What is the typical depth of analysis [of XPS]?

The 2-10 nm depth resolution of XPS measurements is now included in the manuscript in the methods as well as the main text.

References:

Cáceres, F., Wadsworth, F. B., Scheu, B., Colombier, M., Madonna, C., Cimorelli, C., Hess, K. U., Kaliwoda, M., Ruthensteiner, B., & Dingwell, D. B. (2020). Can nanolites enhance eruption explosivity? *Geology*, 48(10), 997–1001. <https://doi.org/10.1130/G47317.1>

Di Genova, D., Brooker, R. A., Mader, H. M., Drewitt, J. W. E., Longo, A., Deubener, J., Neuville, D. R., Fanara, S., Shebanova, O., Anzellini, S., Arzilli, F., Bamber, E. C., Hennem, L., La Spina, G., & Miyajima, N. (2020). In situ observation of nanolite growth in volcanic melt: A driving force for explosive eruptions. *Science Advances*, 6(39), 413–436. <https://doi.org/10.1126/sciadv.abb0413>

Di Genova, D., Zandona, A., & Deubener, J. (2020). Unravelling the effect of nano-

heterogeneity on the viscosity of silicate melts: Implications for glass manufacturing and volcanic eruptions. *Journal of Non-Crystalline Solids*, 545.
<https://doi.org/10.1016/j.jnoncrysol.2020.120248>

- Eychenne, J., Le Pennec, J. L., Troncoso, L., Gouhier, M., & Nedelec, J. M. (2012). Causes and consequences of bimodal grain-size distribution of tephra fall deposited during the August 2006 Tungurahua eruption (Ecuador). *Bulletin of Volcanology*, 74(1), 187–205.
<https://doi.org/10.1007/s00445-011-0517-5>
- Mollo, S., & Hammer, J. E. (2017). Dynamic crystallization in magmas. *European Mineralogical Union Notes in Mineralogy*, 16, 373–418. <https://doi.org/10.1180/EMU-notes.16.12>
- Watson, E. B. (1996). Surface enrichment and trace-element uptake during crystal growth. In *Geochimica et Cosmochimica Acta* (Vol. 60, Issue 24).
- Watson, E. B., & Müller, T. (2009). Non-equilibrium isotopic and elemental fractionation during diffusion-controlled crystal growth under static and dynamic conditions. *Chemical Geology*, 267(3–4), 111–124. <https://doi.org/10.1016/j.chemgeo.2008.10.036>

REVIEWER COMMENTS

Reviewer #3 (Remarks to the Author):

Dear editor,

in this paper, Hornby et al. present a study based on controlled fragmentation experiments on andesitic products, which were sampled from the deposits of the 2006 eruption of Tungurahua volcano. The authors address the dynamics of deflection of fractures during fragmentation, and their effects on the external surface of ash particles in terms of geochemistry and mineralogy. They show that nanoscale melt heterogeneities can control the fragmentation dynamics and fracture patterns, which translates into mineralogical and compositional differences between surface and bulk composition of the ash fragments. The paper is well-written, results are reasonably novel, and conclusions are supported in results. The objectives and the importance of this study are well discussed in the introduction, as well as its implications for the study of explosive eruptions. Even though a few intermediate subtitles would be nice in the introduction (I am not sure if the journal Nature Communications allows them), the general structure of the paper is clear and well-organized. I have some major doubts about the paper:

1. In my opinion, the effective scope of this investigation is not clear in terms of the magma properties and thermodynamics conditions under which nanocrystals are relevant on ash surface properties. I do not intend to say that it is necessary to carry out controlled fragmentation experiments in different melts, but it is at least relevant to constrain (or discuss in detail) the conditions under which Fe-rich nanophases are stable and the kinematics of this process is fast enough to affect the ash surface properties. It is likely controlled by oxygen fugacity and diffusion rate, for instance.
2. On the other hand, it is well known that magma fragmentation can be driven by a series of mechanisms, and the resulting fracture patterns are likely different (also revealed by morphological parameters of ash particles). This is scarcely discussed in the manuscript, and, in my opinion, this should be addressed to better define the scope of the results presented. What fragmentation mechanism are the experiments expected to reproduce? Is this fragmentation experiment expected to be applicable for large-scale Plinian eruptions? (that is, for the relevant subset of eruptions in terms of their effect on climate, discussed in the first paragraph). How the bubble number density and bubble dimensions are expected to affect the fracture pattern? I always ask myself this question when I read about controlled fragmentation experiments, and I believe that in this work it is particularly relevant.
3. Sub-Plinian eruptions are almost intrinsically oscillating events, with an heterogenous conduit (laterally, and likely vertically). Although the authors describe in detail the phase of the eruption for which the samples are expected to be representative (Section Methods: Natural sample collection and

generation), I wonder if, for a given phase of the eruption, portions of the melt that did not fragment finely (I mean, for example, a sample extracted from a 200 cm³ bomb) can be considered equivalent to a melt portion that effectively generated ash (for example, in terms of texture), considering that internal heterogeneities documented by the authors are observable at a much smaller scale. Considering the oscillating nature and internal variability of sub-Plinian products, I have doubts that this type of event is an ideal case for carrying out this type of study.

4. Data presented in the manuscript derive from different analytical methods. Even though the practical reasons for selecting the different methods are clear, in my opinion, the authors must demonstrate better that the likely systematic differences between geochemical results derived from different analytical methods are not the cause of the geochemical trends presented in the paper.

All in all, my suggestion is moderate revision. Below I present some editorial comments and suggestions for different lines of the submitted manuscript PDF.

Additional editorial comments/suggestions:

L25: andesite generates > andesitic melts generate

L26: disparity > variability

L33: , an essential consideration > . Thus, we propose essential considerations

L37: mediator of > factor controlling

L38: reactions > reaction

L39: extracts > extract

L45: concentrations > concentration

L52: parity > equivalence

L55: strength > dimensions, shape and strength

L55: I suggest to include additional simulations.

L68: ballistic > ballistic and fallout.

I mean, if you include ash, this material was probably deposited as pyroclastic fall.

L78: We > Hereafter, we

L82: differences between the two > discrepancies

L116: "Methods" should be an hyperlink

L184: and is > , being

L189: Two strange horizontal lines are present in the text.

L206: glass(Figure > glass (Figure

L260: and not column > and not from column

L266: There is an space between dynamics and 26.

L275: Figure 6 > Figure 6)

L278-288: Part of this Section is redundant. I mean, it is presented in the main text of the manuscript.

L369: Can you provide a sort of statistical parameter to quantify the quality of the calibration?

L370: This line skip is not necessary.

L594: image, s > image

L605: Calculated > calculated

L610: I suggest using a letter to indicate every panel and describe them in the caption using the indications a-b), c), d) and e-g).

L653: P, T > P-T

L698: There is an indentation issue

L698-702: I suggest to separate this information in two sentences.

L821: showsshowing > showing

L864: and show > and exhibit

L866: and 6 > and 6)

L874: microtexturesMicrotextures > Microtextures

L882: Extended Data > Supplementary

L887: August > August 2006

L989: microlite. (circled > microlite (circled

Reviewer #4 (Remarks to the Author):

I read with interest the work by Dr. Hornby and co-workers, but I found it not completely persuasive for the quality required of a Nature Communication paper.

This work could add a relevant contribution to science in the field of geochemistry of volcanic ashes. However, I have some general concerns about the main hypothesis that the Authors seems to challenge in the present work. Additionally, I have some other remarks listed below that should be taken into consideration to improve the overall quality of the paper.

Main comments:

The scientific significance of this work stems from the consideration that “previous investigations on volcanic ash have assumed the theoretical initial surface to be equivalent to the composition of the bulk particle assemblage”. To illustrate this point, the Authors cite two papers, namely:

11. Gislason, S. R. et al. Characterization of Eyjafjallajökull volcanic ash particles and a protocol for rapid risk assessment. *Proc. Natl. Acad. Sci. U. S. A.* 108, 7307–7312 (2011).

12. Delmelle, P., Lambert, M., Dufrêne, Y., Gerin, P. & Óskarsson, N. Gas/aerosol-ash interaction in volcanic plumes: New insights from surface analyses of fine ash particles. *Earth Planet. Sci. Lett.* 259, 159–170 (2007).

However, both papers acknowledge the (strong) discrepancy between the surface and bulk chemistry of the ash. Therefore, these two papers should not be used to support the statement above, and in general, Authors should consider the fact that several previous works acknowledged the discrepancy between surface and bulk composition of volcanic ashes. An opposite statement might be inappropriate for the introduction. Dr. Wright already made a similar remark during the previous round of revision. I might agree with the authors about the fact that this paper suggests "a new appreciation of highly depth-sensitive chemical disparity" in volcanic ashes. However, the authors should try to rephrase their initial hypothesis as it appears to be biased. They should probably focus more on the mechanisms of fragmentation that generate chemically different surfaces (the "nanoscale heterogeneities caused by disequilibrium crystallization and/or unmixing during pre-eruptive undercooling"), which, in my opinion, represent the actual scientific breakthrough of this paper.

To attain size-homogeneity, the Authors sieved fragmented and crushed materials to the same μm -sized fraction (63-90 μm). I see the practical reason to choose a particle size range which is compatible with the imaging and analytical techniques used in the work. I still wonder, however, if there are other scientific reasons that support the design of the study with the adoption of that size range? Specifically, I wonder if selecting a larger or smaller size range would have led to similar results, in terms of surface vs bulk chemistry of particles. Somehow, the Authors seems to suggest that the nanoscale understanding of the different surface chemistry of their experimentally-generated volcanic ashes also applies to particles with smaller sizes, since they claim their work has implications also for human health. However, exposed this size fraction is too large to have a toxicological relevance in humans, which respiratory system efficiently clear particle with diameter $> 10 \mu\text{m}$. In this sense, it would be important to evaluate whether the mechanisms of formation of respirable dusts might be explained by a similar texture and chemically-driven fracturing process.

General improvements and minor remarks:

- In the introduction, Authors stated that “Most magmas are multiphase systems bearing crystal phases that vary in strength¹⁵.” Despite one can grasp phenomenologically the meaning of this sentence, it is not clear what we can consider as the “strength” of a crystal phase. Given the central role of fragmentation in this paper, details should be given about what properties are involved in the resistance to deformation/rupture.

- The Authors use notation atm.% to indicate the atomic percent, i.e., the percentage of one kind of atom relative to the total number of atoms in the mineral formula. I am keen to consider at.% the correct notation to express these values.

- Figure 2 reports about the comparative surface and bulk compositions of the fragmented and crushed clasts. Despite this figure is the “heart” of the paper, it is not sufficiently clear and informative. Specifically:

-- Data should be discussed with the support of a proper statistical analysis to determine whether the differences observed are statistically significant.

-- Why a ratio equal to 0.6 is reported on the y axis as the expect “neutral value” for all graphs? How can a ratio of 0.6 might indicate the ratio between XPS and XRF at.% that signal an identical elemental abundance at the surface and in the bulk?

-- Even more puzzling, how can 0.6 be the ratio that indicates relative variation of the major crystalline phases between the bulk and the micron-scale surface of particles?

- On page 7, Authors discuss Figure 3, panel d-e. However, no panel “e” is visible in Figure 3, nor it is mentioned in the caption.

- Figure 3a, scale bars are not reported. If panel b and c are magnifications of a portion of panel a, please highlight them of the figure.

- Is actually Figure 3d a secondary electron SEM image? The bright spots and the blurred 3D features make me think more about a backscattered electron image.

- Are color maps in Figure 3 (bottom right) the false-color X-ray elemental mapping of SEM image in panel d (bottom left)? Please highlight the mapped zone in panel d, if this is the case.

REVIEWER COMMENTS

Reviewer #3 (Remarks to the Author):

Dear editor,

in this paper, Hornby et al. present a study based on controlled fragmentation experiments on andesitic products, which were sampled from the deposits of the 2006 eruption of Tungurahua volcano. The authors address the dynamics of deflection of fractures during fragmentation, and their effects on the external surface of ash particles in terms of geochemistry and mineralogy. They show that nanoscale melt heterogeneities can control the fragmentation dynamics and fracture patterns, which translates into mineralogical and compositional differences between surface and bulk composition of the ash fragments. The paper is well-written, results are reasonably novel, and conclusions are supported in results. The objectives and the importance of this study are well discussed in the introduction, as well as its implications for the study of explosive eruptions. Even though a few intermediate subtitles would be nice in the introduction (I am not sure if the journal Nature Communications allows them), the general structure of the paper is clear and well-organized. I have some major doubts about the paper:

1. In my opinion, the effective scope of this investigation is not clear in terms of the magma properties and thermodynamics conditions under which nanocrystals are relevant on ash surface properties. I do not intend to say that it is necessary to carry out controlled fragmentation experiments in different melts, but it is at least relevant to constrain (or discuss in detail) the conditions under which Fe-rich nanophases are stable and the kinematics of this process is fast enough to affect the ash surface properties. It is likely controlled by oxygen fugacity and diffusion rate, for instance.

We have added a paragraph, copied below, describing the oxygen fugacity conditions, timescales, tectonic settings and eruption scenarios for which we expect nanolites or SLI to be relevant during magma fragmentation. With this, we include a number of recent examples in the volcanology literature, which has seen an enormous uptick in papers dealing with nanotextures in the past 5 years (see Scarani et al. 2022, Figure 1). Further, we now elaborate on the examples and conditions for formation in the Supplementary Discussion. We also want to emphasize that, while the data presented are relevant for a now-defined set of conditions, the scope of the finding (that microtextures influence surface chemistry) is likely far more widely applicable. Consequently, we have rephrased parts of the manuscript to make this abundantly clear.

We include the following paragraph in the main manuscript (lines 214-222):

“Here, we define a model of ash surface creation modulated by rapid pre-eruptive growth of mafic crystals and the presence or development of a nanoscale Fe-rich phase in the matrix glass. These features have been documented in recent explosive products from mafic (Stromboli(1) and Mt Etna including the 122 BC Plinian eruption(2)) through silicic (Havre(3)) explosive eruptions. The conditions for formation of the set of melt nanoheterogeneities that we document here (described in detail in Supplementary Discussion 4), are most applicable to relatively oxidized arc environments, particularly those with higher Fe and alkali content, and for magma systems perched

close to a critical threshold where mafic recharge may cause disequilibrium crystallization or unmixing and lead to sharp viscosity increases and explosive eruption(2, 4).”

We have included the following In Supplementary Discussion 4

“Natural conditions for development of Fe-Ti melt nanotextures

The development of Fe(Ti) nanolites is controlled by the oxidation state and coordination of Fe (determined by the oxygen fugacity(5)) the concentration of Fe(*I*) in a silicate melt and the undercooling. The injection of relatively mafic, hot and volatile-rich magma into a more evolved and cooler magma body (known as mafic recharge) is established as a common triggering process for volcanic eruptions(6, 7), including at Tungurahua(8, 9), and a potential source for oxidizing fluids (particularly water) that increase oxygen fugacity and generate high ΔT that may trigger nanolite crystallization(10) and silicate liquid immiscibility in andesitic magmas(11, 12). Oxygen fugacity conditions increasing from QFM+1 to QFM+2 are shown to favour nanolite crystallization(10); a range of QFM+1.5-3 in a high-Fe andesite is modelled to promote silicate liquid immiscibility(13). These conditions are in the range for arc tectonic settings, although not for plume or mid-ocean ridge volcanism(14). For nanolites, timescale of nanolite formation are highly dependent on cooling rate, but vary from >1000 minutes (>17 hours) for rhyolite(15) to the first 100s of seconds of cooling for basaltic compositions at high cooling rates(2, 16). For SLI, in the case of binodal CBL-triggered unmixing, timescales may be governed by diffusion rates in the melt(17) with unmixing textures maturing over timescales likely longer than for nanolite formation.

In addition to the basaltic and rhyolitic examples in the main text, the nanotextures documented in this study have been recently recorded in andesitic (Shinmoedake(18)), trachyandesitic (Tambora(2)), and trachytic (Fukutoku-oka-no-Ba(10)) volcanic eruption products.”

2. On the other hand, it is well known that magma fragmentation can be driven by a series of mechanisms, and the resulting fracture patterns are likely different (also revealed by morphological parameters of ash particles). This is scarcely discussed in the manuscript, and, in my opinion, this should be addressed to better define the scope of the results presented. What fragmentation mechanism are the experiments expected to reproduce? Is this fragmentation experiment expected to be applicable for large-scale Plinian eruptions? (that is, for the relevant subset of eruptions in terms of their effect on climate, discussed in the first paragraph). How the bubble number density and bubble dimensions are expected to affect the fracture pattern? I always ask myself this question when I read about controlled fragmentation experiments, and I believe that in this work it is particularly relevant.

This an excellent point by the reviewer and we have taken care to integrate this comment throughout the manuscript through both additions and deletions.

Rapid decompression experiments are most relevant for collapse and magmatic unloading events in volcanism. These commonly occur for lava domes (e.g. Soufriere Hills, 2003 – Herd et al. 2005), but are

also relevant for sector collapse eruptions (e.g., Mt St Helens, 1980, Santa Maria 1902) and Vulcanian explosions. The other type of fragmentation that we investigate here is direct impact, which is comparable to secondary fragmentation processes in PDCs and any eruptive column with a significant range in particle velocity or direction. We see the same patterns in chemical variation at particle surfaces for these two fragmentation modes.

For Plinian eruptions, bubble expansion and shear are typically the dominant drivers of fragmentation, although rapid decompression may trigger or accompany the eruption. Given that these dominant fragmentation modes are different to the two modes investigated here, it is possible that the sensitivity of fractures to nanotextures during fragmentation during Plinian eruptions is different.

However, the principles we have described here, that melt nano-heterogeneities can affect fracture paths during fragmentation are likely to apply to some extent to all but the most energetic scenarios, where fragmentation energy greatly outstrips material sensitivity (e.g. in phreatomagmatic eruptions).

We introduced climate-changing eruptions as a general point on the importance of ash surface chemistry, however we have deleted the mention in the introduction, and included a summary sentence of the above discussion, given below, in lines 223-240.

“However, a broader range of heterogeneous melt textures generated during the disequilibrium conditions that commonly accompany or trigger magma ascent, shallow storage and eruption are likely to affect magma fragmentation paths and subsequent ash surface chemistry. The experimental fragmentation mechanisms in this study reproduce natural fragmentation triggered by dome or edifice collapse or unloading events (rapid decompression) including plugged Vulcanian-style eruptions and secondary fragmentation by collisions between pyroclasts in the plume or in PDCs. Such eruptive events and phenomena are frequently observed at explosive volcanoes with intermediate magma compositions in subduction settings. These are the most frequently erupting volcanoes(19), and hence are the dominant sources of volcanic ash on a global scale (e.g., >70 wt.% of ash + PDC deposits from VEI 1-5 eruptions in subduction settings during the last 40 years(20)). Although we cannot claim that the rapid decompression experiments well-reproduce the fragmentation process of the erupted material during the sub-Plinian phase of the Tungurahua 2006 eruption, the crushed experimental samples better simulate secondary fragmentation by collisions between pyroclasts in PDCs²⁰, which is the mechanism that has been proposed to fragment the natural ash bearing the nanotextures described here (see Methods, Figure 2e-g, Supplementary Figure 5).

Within these limits, our experiments suggest that the sensitivity to nanotextures would apply to all but the most energetic fragmentation modes, for example, phreatomagmatic and Plinian strain-induced fragmentation⁴⁷, where further investigation is required. Excluding such high-energy eruption modes, we can conclude that the surface chemistry of ash from eruptive events, and by extension, the reactivity of that surface, may differ from that implied by a micro- or bulk scale analysis.”

3. Sub-Plinian eruptions are almost intrinsically oscillating events, with an heterogeneous conduit (laterally, and likely vertically). Although the authors describe in detail the phase of the eruption for which the samples are expected to be representative (Section Methods: Natural sample collection and generation),

I wonder if, for a given phase of the eruption, portions of the melt that did not fragment finely (I mean, for example, a sample extracted from a 200 cm³ bomb) can be considered equivalent to a melt portion that effectively generated ash (for example, in terms of texture), considering that internal heterogeneities documented by the authors are observable at a much smaller scale. Considering the oscillating nature and internal variability of sub-Plinian products, I have doubts that this type of event is an ideal case for carrying out this type of study.

Our main motivation for our choice of case study was the representativity of such eruption types at the global scale. VEI 2-3 eruptions at andesitic volcanoes are the most frequent explosive eruptions of the global record (19, 20), with recognized impacts of volcanic ash on the environment, society and agriculture (21–23). We would argue that all eruptions, including higher intensity eruptions such as Plinian events, show temporal variations in eruptive dynamics (demonstrated by complex tephra sequences, e.g. (24)), as well as variabilities in fragmentation efficiencies (evidenced by the concomitant eruption of bombs several cm in size and volcanic ash finer than 100 μm).

In the Methods, we describe evidence that bombs produced in the eruption that are similar to the bomb studied here, were most likely milled and abraded in PDCs to produce the natural co-PDC ash, rather than fragmenting during ejection from the vent in fire-fountaining activity. Hence, we are not claiming equivalency to the ash produced during the sub-Plinian phase III or that our results are applicable to all the eruptive products, as made clear by our estimation of the 15-20% of natural ash samples bearing the same nano-textures as our experimental materials.

Although we cannot claim that the rapid decompression experiments well-reproduce the fragmentation process of the erupted material during the sub-Plinian phase, in other eruptive scenarios at intermediate to silicic arc volcanoes a similar magmatic mixing, ascent and shallow storage history may precede lava dome destabilization and collapse or plug pressurization and release (Vulcanian eruption) leading to a major eruption driven by rapid decompression.

On the other hand, a fragmentation mode producing ash from the starting material during the 16-17th August eruption (e.g., PDC milling) is reproduced by the direct impact or crushing experiments. Material abrasion in PDCs and formation of co-PDC plumes transporting the ash produced by these abrasion processes occur systematically when PDCs are generated during explosive eruptions (25). In that respect, our study has a broad representativity.

There are undoubtedly eruptions where the observed nanotextures and experimental fragmentation mechanisms would have been more applicable to the bulk fragmented products; but we reiterate the relevance of the principle of our study—i.e., since this is a pioneering study, investigation into the effects of other melt (nano)textures (such as diffusion layers around bubbles, melt mixing/mingling, vestiges of foam collapse) on fracture pathways are not established; but our study suggests mechanisms by which they may have a similar effect. Further work is certainly required to better understand such effects, and our study provides a framework for this.

Consideration of these important points raised by the reviewer contributed to the new paragraph given in the answer to the previous point and in strengthening our descriptions of the rationale for this study.

4. Data presented in the manuscript derive from different analytical methods. Even though the practical

reasons for selecting the different methods are clear, in my opinion, the authors must demonstrate better that the likely systematic differences between geochemical results derived from different analytical methods are not the cause of the geochemical trends presented in the paper.

We use four analytical techniques in the paper: micro-XRF, SEM-EDS and BSE (QEMSCAN), EPMA and XPS to produce three complementary datasets involving different resolution: micro-XRF (bulk chemistry), QEMSCAN+EPMA (microscale surface and bulk chemistry), and XPS (nanoscale surface chemistry). The average results for each sample and each technique (bulk QEMSCAN+EPMA chemistry) are shown in a new Supplementary Figure 8, shown below.

We note some observations from these comparative results.

1. QEMSCAN+EPMA and micro-XRF measurements are in good agreement for all elements
2. Maximum differences between XPS and the other measurements modes for Ca and Al are less than 20%, whereas for Fe and Na differences are up to ~180% and for Mg reach ~800%.
3. The zig-zag patterns of relatively increasing or decreasing concentration between samples shown below are reproduced in both XPS and QEMSCAN+EPMA data for Al, Ca, Fe, and K, but not for Mg and Na. For these latter two elements, the XPS data shows the same relative concentration patterns as for K, but the EPMA+QEMSCAN data are relatively flat.

The similarity of the results in terms of absolute concentration (Al and Ca) and relative concentration between crushed and shock tube samples (Al, Ca and K) shows that a systematic bias is not present between the different measurement techniques. To our knowledge, biases affecting Mg, Fe, and Na (points 2 and 3) in SEM-EDS+EPMA are the well-known glass measurement issues of Na-mobility (and time-dependent depletion in measurements) and Fe oxidation state uncertainty, however we estimated glass chemistry by difference instead of relying on EPMA measurements (see Methods).

EPMA chemistry determination and SEM-based phase mapping are discipline standards in Earth science and the mining industry, respectively, and synthesis of these data types is common in the literature. XPS is a newer technique, but has been increasingly employed in the Earth sciences and other fields in the past decades, and comparisons between nanoscale surfaces measured by XPS and bulk compositions measured by other techniques have been made in numerous studies on volcanic products (26, 27, 36, 28–35).

We have included all of the above discussion into a new Supplementary Discussion 5.

Supplementary Figure 8. Comparison of average elemental concentration from XPS (yellow squares) and micro-XRF (blue dots) measurements, and from calculations based on combined EPMA and QEMSCAN measurements (red triangles).

XPS measurement quantification is sensitive to peak deconvolution methods and data analysis – e.g. (37). We took the following steps to ensure XPS data integrity, which is now included in the manuscript as Supplementary Discussion 6.

“XPS quantification is based on the measurement of peak areas above the spectrum background. The intensity or area (A) of the peaks, depends on the photon flux (J), the concentration of the atom/ion in the solid (ρ), the cross-section (σ) for photoelectron emission (which depends on the element and energy being considered), instrumental factors (K), and the electron attenuation length (λ), ($A = J \cdot \rho \cdot \sigma \cdot K \cdot \lambda$). In practice, atomic sensitivity factors (F) containing J , σ , K , λ , are used, and the quantification is given as the fraction of each peak area divided by the sensitivity factor, normalised for all peaks. [i] atomic % = $\{(A_i/F_i)/\Sigma(A/F)\}$ where A_i and F_i are the peak area and sensitivity factor of element i . and $\Sigma(A/F)$ is the sum of the peak area/sensitivity factor ratios for all elements. Therefore, the measurement uncertainty basically springs from how well the peak area is measured, i.e. how well the peak is acquired (high signal to noise ratio is needed) and how well the background is subtracted. For data analysis we used the software casaXPS (SINTEF's licence) and for the quantification we used the whole spectrum area i.e. no peak deconvolution was performed. Therefore, quantification was not dependent on peak fitting something that contains a significant degree of uncertainty.

To make sure that no errors were introduced due to instrumental factors all samples were measured on the same instrument and the same vacuum level. The spectra were acquired at the same angle of emission (0° , vertical emission) and the same analyser acceptance angle. In all samples, each peak corresponding to a specific element was acquired at the same acquisition time and the same background type was subtracted for the same peak in all samples. For the quantification we used the instrument provider Wagner sensitivity factors stored in the spectrometer's library. These are empirical and very reliable as are based on measurements performed on standards. The same x-ray power was used for all measurements, all high-resolution peaks were acquired at the same pass energy. For the quantification we measured areas of peaks having the same energy to eliminate discrepancies arising from attenuation length differences. As an example, we compared the Mg 2p peaks in all samples and not Mg 1s in one sample with the Mg 2p in another sample. The Mg 1s photoelectron has lower attenuation length and its signal originates from the outermost surface, whilst the Mg 2p has higher attenuation length and originates from slightly deeper in the sample. Thus, we compare the elemental content at the same depth for all samples. However, the analysis depth of XPS of <10 nm is 2-3 orders of magnitude smaller than that of XRF/EDS/EPMA and this difference in scale makes the comparison between surface and bulk evident.

To ensure that no errors attributed to the design of the experiment were introduced in the XPS measurements, the following measures were taken. We analysed 5 different areas of each sample. These gave identical spectra ensuring measurement reproducibility. Each sample was tested for irradiation induced diffusion of alkaline elements (K, Na). For this we performed time resolved experiments by acquiring survey spectra at different acquisition times 1, 5, 10, minutes. The results showed no such dependence as the spectra exhibited the same peak intensity ratios. To ensure absence of irradiation effects throughout the whole measurement duration for each sample, spectra acquisition started and ended with acquiring survey spectra with the same acquisition parameters. The comparison showed no irradiation effects as the peak intensity ratios of the first and last survey spectrum of the same sample were identical.”

Finally, we added the following into Supplementary Discussion 1:

“Direct comparison of the bulk and nanoscale surface measurements in Supplementary Figure 8 suggest that the depth dependence for concentration (i.e., the difference between XPS measurements and microscale or bulk measurements) and the sensitivity to fragmentation mode varies for different elements. The most depth-dependent elemental concentrations appear to be Mg, K, Na and Fe, in that order. These elements also show variations in concentration in the EDS-linescans between compositional boundary layers and microlite-bearing glass in Supplementary Figure 3. A dependence on fragmentation mode is found for all elements, but is most pronounced and consistent for Mg, Na and K where, in all cases, crushed samples have higher concentration at the nm-scale than samples fragmented by rapid decompression under the same conditions. This pattern is also found at the micron scale for K, but not Mg or Na (see Figure 2a, c and Supplementary Figure 8).”

All in all, my suggestion is moderate revision. Below I present some editorial comments and suggestions for different lines of the submitted manuscript PDF.

Additional editorial comments/suggestions:

L25: andesite generates > andesitic melts generate

We have to disagree with this suggestion. We fragmented natural material and without solid phases, none of our studies results are relevant.

L26: disparity > variability

Disparity = unequal, which is our intended meaning, therefore we opt not to change this.

L33: , an essential consideration > . Thus, we propose essential considerations

This was a long sentence and we have deleted the preceding clause to make it more clear. It now reads:

“In this manner, we argue that commonly-observed pre-eruptive microtextures can generate primary discrepancies in ash surface chemistry, an essential consideration for understanding the cascading consequences of reactive ash surfaces in various environments.”

L37: mediator of > factor controlling

We prefer mediator here, it is a more specific description – i.e., the ash surface is the interface between volcanic ash and the environment and determines the short-term reactivity.

L38: reactions > reaction

Changed as suggested

L39: extracts > extract

Changed as suggested

L45: concentrations > concentration

Changed as suggested

L52: parity > equivalence

These words have the same meaning, but we have changed as suggested.

L55: strength > dimensions, shape and strength

Changed as suggested

L55: I suggest to include additional simulations.

We are not sure what is intended here?

L68: ballistic > ballistic and fallout.

I mean, if you include ash, this material was probably deposited as pyroclastic fall.

True, changed as suggested.

L78: We > Hereafter, we

Changed as suggested.

L82: differences between the two > discrepancies

Changed as suggested.

L116: "Methods" should be an hyperlink

We have not included links in the draft.

L184: and is > , being

Changed as suggested

L189: Two strange horizontal lines are present in the text.

Now deleted

L206: glass(Figure > glass (Figure

Corrected

L260: and not column > and not from column

Corrected

L266: There is an space between dynamics and 26.

Corrected

L275: Figure 6 > Figure 6)

Corrected

L278-288: Part of this Section is redundant. I mean, it is presented in the main text of the manuscript.

Some repetition here is unavoidable to provide a coherent narrative in the main text. We have limited to the extent we believe possible.

L369: Can you provide a sort of statistical parameter to quantify the quality of the calibration?

Unfortunately, we do not have this available.

L370: This line skip is not necessary.

Corrected.

L594: image, s > image

Corrected.

L605: Calculated > calculated

Corrected.

L610: I suggest using a letter to indicate every panel and describe them in the caption using the indications a-b), c), d) and e-g).

Thank you, changed as suggested.

L653: P, T > P-T

Corrected.

L698: There is an indentation issue

Corrected.

L698-702: I suggest to separate this information in two sentences.

Thanks, this was a long sentence. Changed and elaborated slightly.

L821: showshowing > showing

Corrected.

L864: and show > and exhibit

Corrected.

L866: and 6 > and 6)

Corrected.

L874: microtexturesMicrotextures > Microtextures

Corrected.

L882: Extended Data > Supplementary

Corrected.

L887: August > August 2006

Corrected.

L989: microlite. (circled > microlite (circled

Corrected.

Reviewer #4 (Remarks to the Author):

I read with interest the work by Dr. Hornby and co-workers, but I found it not completely persuasive for the quality required of a Nature Communication paper.

This work could add a relevant contribution to science in the field of geochemistry of volcanic ashes.

However, I have some general concerns about the main hypothesis that the Authors seems to challenge in the present work. Additionally, I have some other remarks listed below that should be taken into consideration to improve the overall quality of the paper.

Main comments:

The scientific significance of this work stems from the consideration that “previous investigations on

volcanic ash have assumed the theoretical initial surface to be equivalent to the composition of the bulk particle assemblage". To illustrate this point, the Authors cite two papers, namely:

11. Gislason, S. R. et al. Characterization of Eyjafjallajökull volcanic ash particles and a protocol for rapid risk assessment. *Proc. Natl. Acad. Sci. U. S. A.* 108, 7307–7312 (2011).

12. Delmelle, P., Lambert, M., Dufrêne, Y., Gerin, P. & Óskarsson, N. Gas/aerosol-ash interaction in volcanic plumes: New insights from surface analyses of fine ash particles. *Earth Planet. Sci. Lett.* 259, 159–170 (2007).

However, both papers acknowledge the (strong) discrepancy between the surface and bulk chemistry of the ash. Therefore, these two papers should not be used to support the statement above, and in general, Authors should consider the fact that several previous works acknowledged the discrepancy between surface and bulk composition of volcanic ashes. An opposite statement might be inappropriate for the introduction. Dr. Wright already made a similar remark during the previous round of revision. I might agree with the authors about the fact that this paper suggests "a new appreciation of highly depth-sensitive chemical disparity" in volcanic ashes. However, the authors should try to rephrase their initial hypothesis as it appears to be biased. They should probably focus more on the mechanisms of fragmentation that generate chemically different surfaces (the "nanoscale heterogeneities caused by disequilibrium crystallization and/or unmixing during pre-eruptive undercooling"), which, in my opinion, represent the actual scientific breakthrough of this paper.

We are not claiming that the authors did not find differences between surface and bulk chemistry – we are only disputing their inference that the differences they found were entirely generated from gas-aerosol interaction and were not present beforehand (e.g., generated during the fragmentation process).

However, we agree that the way this was phrased was not entirely clear and reads as more combative than necessary. We have toned down the criticism of previous studies and rephrased the relevant sentence in the introduction (lines 46-50) to read:

“Previous studies^{11,12} have invoked alteration by various in-plume and post-eruptive processes to explain differences between the nanoscale surface and bulk chemistry of the ash particles. However, no previous study has considered whether such discrepancies could derive from an *a priori* heterogeneity created from the chemistry, texture and fragmentation mode(s) of crystal-bearing magmas during fragmentation(38, 39).”

We have now included substantially more detail on the applicability of the fragmentation mechanisms and nanoscale textures investigated in our study (e.g., lines 220-225, 230-240 and Supplementary Discussion 4). We include a new sentence in the introduction to highlight that the texture-driven mechanisms for surface chemistry discrepancy we explore are inferred to be broadly applicable during volcanic fragmentation (lines 59-66):

“The volcanic materials were generated in the VEI 3 eruption of Tungurahua volcano, Ecuador(9) on August 16-17th 2006. We do not argue here for universal applicability of the models outlined here to all volcanic surfaces, but rather use them to underpin an essential argument: if unique ash surfaces are created from combinations of magmatic, textural and

eruptive conditions in one situation, then there may be a wide array of eruptive surfaces created across the spectrum of eruptive events. Any future studies of ash surface-mediated reactions may be compromised in their utility, as long as this subject area remains unexplored.”

...which we reiterate in our concluding sentences in the manuscript (lines 241-249):

“We emphasise that the model outlined for the Tungurahua materials in this study is not to be universally applied. However, the foundational principle put forward by this study, that nanoscale surface chemistry of ash surfaces is likely strongly influenced by the textural, magmatic and fragmentation conditions, should be considered in all cases. We advocate for detailed observation and measurements of surface and near-surface features, together with targeted experimental and analytical work with a focus on discerning the influence of textural and fragmentation controls on those surfaces. Such steps are to accurately (re)assess the host of ash surface-mediated reactions and impacts to the atmosphere and depositional environments.”

To attain size-homogeneity, the Authors sieved fragmented and crushed materials to the same um-sized fraction (63-90 μm). I see the practical reason to choose a particle size range which is compatible with the imaging and analytical techniques used in the work. I still wonder, however, if there are other scientific reasons that support the design of the study with the adoption of that size range? Specifically, I wonder if selecting a larger or smaller size range would have led to similar results, in terms of surface vs bulk chemistry of particles. Somehow, the Authors seems to suggest that the nanoscale understanding of the different surface chemistry of their experimentally-generated volcanic ashes also applies to particles with smaller sizes, since they claim their work has implications also for human health. However, exposed this size fraction is too large to have a toxicological relevance in humans, which respiratory system efficiently clear particle with diameter > 10 μm . In this sense, it would be important to evaluate whether the mechanisms of formation of respirable dusts might be explained by a similar texture and chemically-driven fracturing process.

The primary fragmentation mechanisms are thought to apply for all particle sizes with rare exceptions (e.g., ultrafine particles produced during rupture in confined experiments on MSH material or in fault gouge during faulting of spine-forming eruptions, (40)).

However, for co-PDC ash generation the dominant mechanism for sub-10 micron ash generation is abrasion – that is, the chipping and rounding of asperities on clasts due to low-energy impacts and friction. The energy involved in fracturing during abrasion is substantially lower than for milling (impact, splitting) and therefore the sensitivity to changes in material properties is higher. We contend that fractures creating sub-10 micron particles during abrasion would show the same heterogeneous surface chemistry if they passed through glass (or melt) containing the nanotextures we describe.

At some point, the scale of the particle and the scale of the nanotextures become comparable (e.g., in the nm diameter range) and the surface and bulk are expected to become congruent, but this is well below the respirable size cutoff.

General improvements and minor remarks:

- In the introduction, Authors stated that “Most magmas are multiphase systems bearing crystal phases that vary in strength¹⁵.” Despite one can grasp phenomenologically the meaning of this sentence, it is

not clear what we can consider as the "strength" of a crystal phase. Given the central role of fragmentation in this paper, details should be given about what properties are involved in the resistance to deformation/rupture.

We changed this sentence to specify that fracture toughness is the most important attribute in this context. It now reads "Most magmas are multiphase systems bearing crystal phases that vary in fracture toughness, or the resistance to crack propagation(41)."

- The Authors use notation atm.% to indicate the atomic percent, i.e., the percentage of one kind of atom relative to the total number of atoms in the mineral formula. I am keen to consider at.% the correct notation to express these values.

We changed to at.% in all instances.

- Figure 2 reports about the comparative surface and bulk compositions of the fragmented and crushed clasts. Despite this figure is the "heart" of the paper, it is not sufficiently clear and informative. Specifically:

-- Data should be discussed with the support of a proper statistical analysis to determine whether the differences observed are statistically significant.

As part of answering this question, the QEMSCAN phase fraction totals and error calculation for the QEMSCAN measurements has been revisited and revised. Specifically, we chose the same minimum size cutoff for particle cores and outlines, which did not consider that outline pixels exceed core pixels for the smallest particles so there were more small particles included in the outline dataset. Phase fractions and standard error have been recalculated by binning random sets of 50 particles with phase fractions weighted by total binned particle area and calculating averages of the means and the standard error. Both the error values and phase fractions are similar to the previous calculations from the entire dataset, but certain phases that were more prevalent in the smallest particles (e.g. 'unclassified') have lower average fractions and error.

We carefully propagated error through the calculations and present the data with bars showing ± 2 standard error (e.g. 95% confidence limits). It is sufficiently clear from figures 2a and 2c that the differences observed for Mg, and in most cases for Na and Fe are significant.

We note that it is impossible to conduct a student's T-test or similar test of significance on the data presented in Figure 2. We present ratios of datasets that were calculated using the averages of multiple sets of data. In a similar way, there is no meaningful way to present data distribution for these ratios.

-- Why a ratio equal to 0.6 is reported on the y axis as the expected "neutral value" for all graphs? How can a ratio of 0.6 might indicate the ratio between XPS and XRF at.% that signal an identical elemental abundance at the surface and in the bulk?

-- Even more puzzling, how can 0.6 be the ratio that indicates relative variation of the major crystalline phases between the bulk and the micron-scale surface of particles?

The ratio is not at 0.6, but at 1:1, as you would expect. This is a logarithmic scale with base 2, therefore the tick marks show increments of 0.2 between 0.2 and 2.

We have now revised these figures using a logarithmic scale starting at 0.1 to avoid any confusion.

- On page 7, Authors discuss Figure 3, panel d-e. However, no panel “e” is visible in Figure 3, nor it is mentioned in the caption.

We have now changed the figure to label each panel, from a) to g) and updated the caption and text accordingly.

- Figure 3a, scale bars are not reported. If panel b and c are magnifications of a portion of panel a, please highlight them of the figure.

Thank you, we have added scale bars to a. Panels b and c are not blown up areas of a, but come from different samples.

- Is actually Figure 3d a secondary electron SEM image? The bright spots and the blurred 3D features make me think more about a backscattered electron image.

This is indeed a BSE image.

- Are color maps in Figure 3 (bottom right) the false-color X-ray elemental mapping of SEM image in panel d (bottom left)? Please highlight the mapped zone in panel d, if this is the case.

They are the false color maps of panel e. They cover a similar area but are actually slightly zoomed out compared to the SE image. We have added frames to the EDX images to show the area in the SE image.

References

1. A. Scarani, A. Zandonà, F. Di Fiore, P. Valdivia, R. Putra, N. Miyajima, H. Bornhöft, A. Vona, J. Deubener, C. Romano, D. Di Genova, A chemical threshold controls nanocrystallization and degassing behaviour in basalt magmas. *Commun. Earth Environ.* **3** (2022), doi:10.1038/s43247-022-00615-2.
2. D. Di Genova, R. A. Brooker, H. M. Mader, J. W. E. Drewitt, A. Longo, J. Deubener, D. R. Neuville, S. Fanara, O. Shebanova, S. Anzellini, F. Arzilli, E. C. Bamber, L. Hennet, G. La Spina, N. Miyajima, In situ observation of nanolite growth in volcanic melt: A driving force for explosive eruptions. *Sci. Adv.* **6**, 413–436 (2020).
3. J. Knafelc, S. E. Bryan, M. W. M. Jones, D. Gust, G. Mallmann, H. E. Cathey, A. J. Berry, E. C. Ferré, D. L. Howard, Havre 2012 pink pumice is evidence of a short-lived, deep-sea, magnetite nanolite-driven explosive eruption. *Commun. Earth Environ.* **3**, 1–11 (2022).
4. F. Cáceres, F. B. Wadsworth, B. Scheu, M. Colombier, C. Madonna, C. Cimarelli, K. U. Hess, M. Kaliwoda, B. Ruthensteiner, D. B. Dingwell, Can nanolites enhance eruption explosivity? *Geology.* **48**, 997–1001 (2020).
5. C. Le Losq, M. R. Cicconi, D. R. Neuville, Iron in silicate glasses and melts: Implications for volcanological processes. *Magma Redox Geochemistry*, 233–253 (2021).

6. S. R. J. Sparks, H. Sigurdsson, L. Wilson, Magma mixing: A mechanism for triggering acid explosive eruptions. *Nature*. **267**, 315–318 (1977).
7. J. S. Pallister, R. P. Hoblitt, A. G. Reyes, A basalt trigger for the 1991 eruptions of Pinatubo volcano? *Nature*. **356**, 426–428 (1992).
8. M. L. Myers, D. J. Geist, M. C. Rowe, K. S. Harpp, P. J. Wallace, J. Dufek, Replenishment of volatile-rich mafic magma into a degassed chamber drives mixing and eruption of Tungurahua volcano. *Bull. Volcanol.* **76**, 1–17 (2014).
9. P. Samaniego, J. L. Le Pennec, C. Robin, S. Hidalgo, Petrological analysis of the pre-eruptive magmatic process prior to the 2006 explosive eruptions at Tungurahua volcano (Ecuador). *J. Volcanol. Geotherm. Res.* **199**, 69–84 (2011).
10. K. Yoshida, A. Miyake, S. H. Okumura, H. Ishibashi, S. Okumura, A. Okamoto, Y. Niwa, M. Kimura, T. Sato, Y. Tamura, S. Ono, Oxidation-induced nanolite crystallization triggered the 2021 eruption of Fukutoku-Oka-no-Ba, Japan. *Sci. Rep.* **13**, 1–9 (2023).
11. A. H. Clark, Fe-Ti-P Oxide Melts Generated through Magma Mixing in the Antauta Subvolcanic Center, Peru: Implications for the Origin of Nelsonite and Iron Oxide-Dominated Hydrothermal Deposits. *Econ. Geol.* **99**, 377–395 (2004).
12. T. Hou, B. Charlier, F. Holtz, I. Veksler, Z. Zhang, R. Thomas, O. Namur, Immiscible hydrous Fe-Ca-P melt and the origin of iron oxide-apatite ore deposits. *Nat. Commun.* **9**, 1–8 (2018).
13. T. Keller, F. Tornos, J. M. Hanchar, D. K. Pietruszka, A. Soldati, D. B. Dingwell, J. Suckale, Genetic model of the El Laco magnetite-apatite deposits by extrusion of iron-rich melt. *Nat. Commun.* **13**, 1–14 (2022).
14. E. Cottrell, S. K. Birner, M. Brounce, F. A. Davis, L. E. Waters, K. A. Kelley, "Oxygen Fugacity Across Tectonic Settings" in *Redox variables and mechanisms in magmatism and volcanism*, D. R. Neuville, R. Moretti, Eds. (AGU Geophysical Monograph, 2021), pp. 33–61.
15. F. Cáceres, B. Scheu, K. U. Hess, C. Cimarelli, J. Vasseur, M. Kaliwoda, D. B. Dingwell, From melt to crystals: The effects of cooling on Fe[Ti] oxide nanolites crystallisation and melt polymerisation at oxidising conditions. *Chem. Geol.* **563** (2021), doi:10.1016/j.chemgeo.2021.120057.
16. F. Arzilli, G. La Spina, M. R. Burton, M. Polacci, N. Le Gall, M. E. Hartley, D. Di Genova, B. Cai, N. T. Vo, E. C. Bamber, S. Nonni, R. Atwood, E. W. Llewellyn, R. A. Brooker, H. M. Mader, P. D. Lee, Magma fragmentation in highly explosive basaltic eruptions induced by rapid crystallization. *Nat. Geosci.* **12**, 1023–1028 (2019).
17. V. C. Honour, M. B. Holness, B. Charlier, S. C. Piazzolo, O. Namur, T. J. Prosa, I. Martin, R. T. Helz, J. Maclennan, M. M. Jean, Compositional boundary layers trigger liquid unmixing in a basaltic crystal mush. *Nat. Commun.* **10** (2019), doi:10.1038/s41467-019-12694-5.
18. K. Matsumoto, N. Geshi, Shallow crystallization of eruptive magma inferred from volcanic ash microtextures: a case study of the 2018 eruption of Shinmoedake volcano, Japan. *Bull. Volcanol.* **83** (2021), doi:10.1007/s00445-021-01451-6.
19. M. J. Farner, C. T. A. Lee, Effects of crustal thickness on magmatic differentiation in subduction zone volcanism: A global study. *Earth Planet. Sci. Lett.* **470**, 96–107 (2017).

20. F. Galetto, M. E. Pritchard, A. J. Hornby, E. Gazel, N. M. Mahowald, Spatial and Temporal Quantification of Subaerial Volcanism From 1980 to 2019: Solid Products, Masses, and Average Eruptive Rates. *Rev. Geophys.* **61** (2023), doi:10.1029/2022RG000783.
21. R. Few, M. T. Armijos, J. Barclay, Living with Volcan Tungurahua: The dynamics of vulnerability during prolonged volcanic activity. *Geoforum.* **80**, 72–81 (2017).
22. N. Ligot, A. G. Caiquetan, P. Delmelle, Drivers of crop impacts from tephra fallout: insights from interviews with farming communities around Tungurahua volcano, Ecuador. *Volcanica.* **5**, 163–181 (2022).
23. L. F. Guimarães, A. Nieto-Torres, C. Bonadonna, C. Frischknecht, A New Inclusive Volcanic Risk Ranking, Part 2: Application to Latin America. *Front. Earth Sci.* **9**, 1–24 (2021).
24. J. L. Arce, J. L. Macías, L. Vázquez-Selem, The 10.5 ka Plinian eruption of Nevado de Toluca volcano, Mexico: Stratigraphy and hazard implications. *Bull. Geol. Soc. Am.* **115**, 230–248 (2003).
25. S. Engwell, J. Eychenne, "Contribution of Fine Ash to the Atmosphere From Plumes Associated With Pyroclastic Density Currents" in *Volcanic Ash: Hazard Observation* (Elsevier Inc., 2016), pp. 68–85.
26. G. Barone, P. Mazzoleni, R. A. Corsaro, P. Costagliola, F. Di Benedetto, E. Ciliberto, D. Gimeno, C. Bongiorno, C. Spinella, Nanoscale surface modification of Mt. Etna volcanic ashes. *Geochim. Cosmochim. Acta.* **174**, 70–84 (2016).
27. G. Barone, E. Ciliberto, P. Costagliola, P. Mazzoleni, X-ray photoelectron spectroscopy of Mt. Etna volcanic ashes. *Surf. Interface Anal.* **46**, 847–850 (2014).
28. D. A. Cristaldi, C. G. Fortuna, A. Gulino, A photoelectron spectroscopy study of lava stones. *Anal. Methods.* **5**, 3458–3462 (2013).
29. P. Delmelle, M. Lambert, Y. Dufrière, P. Gerin, N. Óskarsson, Gas/aerosol-ash interaction in volcanic plumes: New insights from surface analyses of fine ash particles. *Earth Planet. Sci. Lett.* **259**, 159–170 (2007).
30. A. J. Durant, G. Villarosa, W. I. Rose, P. Delmelle, A. J. Prata, J. G. Viramonte, Long-range volcanic ash transport and fallout during the 2008 eruption of Chaitén volcano, Chile. *Phys. Chem. Earth.* **45–46**, 50–64 (2012).
31. S. R. Gislason, T. Hassenkam, S. Nedel, N. Bovet, E. S. Eiriksdottir, H. A. Alfredsson, C. P. Hem, Z. I. Balogh, K. Dideriksen, N. Óskarsson, B. Sigfusson, G. Larsen, S. L. S. Stipp, Characterization of Eyjafjallajökull volcanic ash particles and a protocol for rapid risk assessment. *Proc. Natl. Acad. Sci. U. S. A.* **108**, 7307–7312 (2011).
32. G. Berger, A. Cathala, S. Fabre, A. Y. Borisova, A. Pages, T. Aigouy, J. Esvan, P. Pinet, Experimental exploration of volcanic rocks-atmosphere interaction under Venus surface conditions. *Icarus.* **329**, 8–23 (2019).
33. J. Olsson, S. L. S. Stipp, K. N. Dalby, S. R. Gislason, Rapid release of metal salts and nutrients from the 2011 Grímsvötn, Iceland volcanic ash. *Geochim. Cosmochim. Acta.* **123**, 134–149 (2013).
34. A. Vogel, S. Diplas, A. J. Durant, A. S. Azar, M. F. Sunding, W. I. Rose, A. Sytchkova, C. Bonadonna, K. Krüger, A. Stohl, Reference data set of volcanic ash physicochemical and optical properties. *J.*

- Geophys. Res. Atmos.* **122**, 9485–9514 (2017).
35. A. F. White, M. F. Hochella, Surface chemistry associated with the cooling and subaerial weathering of recent basalt flows. *Geochim. Cosmochim. Acta.* **56**, 3711–3721 (1992).
 36. P. Ayris, P. Delmelle, Volcanic and atmospheric controls on ash iron solubility: A review. *Phys. Chem. Earth.* **45–46**, 103–112 (2012).
 37. D. R. Baer, J. F. Watts, A. Herrera-Gomez, K. J. Gaskell, Evolving efforts to maintain and improve XPS analysis quality in an era of increasingly diverse uses and users. *Surf. Interface Anal.* **55**, 480–488 (2023).
 38. Y. Lavallée, J. E. Kendrick, "A review of the physical and mechanical properties of volcanic rocks and magmas in the brittle and ductile regimes" in *Forecasting and Planning for Volcanic Hazards, Risks, and Disasters* (Elsevier, 2021), pp. 153–238.
 39. J. Taddeucci, C. Cimarelli, M. A. Alatorre-Ibargüengoitia, H. Delgado-Granados, D. Andronico, E. Del Bello, P. Scarlato, F. Di Stefano, Fracturing and healing of basaltic magmas during explosive volcanic eruptions. *Nat. Geosci.* **14**, 248–254 (2021).
 40. L. A. Kennedy, J. K. Russell, Cataclastic production of volcanic ash at Mount Saint Helens. *Phys. Chem. Earth.* **45–46**, 40–49 (2012).
 41. J. G. Spray, Frictional melting processes in planetary materials: From hypervelocity impact to earthquakes. *Annu. Rev. Earth Planet. Sci.* **38**, 221–254 (2010).

REVIEWERS' COMMENTS

Reviewer #3 (Remarks to the Author):

Dear editor,

in this contribution, Hornby et al. address the influence of Fe-rich nanophases on fracture dynamics of pyroclasts during magmatic fragmentation and on the surface chemistry of the resulting ash, which is significantly different from the bulk composition of pyroclasts. The study is based on controlled fragmentation experiments on andesitic products, which derive from the 2006 sub-Plinian eruption of Tungurahua volcano. The paper is well-written and, in general, well-organized. The introduction is self-consistent in terms of references, description of the problem addressed in the text and explanation of the scope of the paper, while conclusions are effectively based on results. The authors addressed successfully the four major comments presented in my previous review, or at least they included additional discussions about some limitations of their results. I acknowledge particularly the modifications associated with my previous comments 2 and 4, related to the dynamics of magma fragmentation in controlled experiments and the eventual systematic discrepancies between pyroclasts compositional results derived from different measurement methods. I have two general, additional comments and a series of editorial comments and suggestions are presented at the end of this text (please note that my mother-tongue is not English).

1) The authors included a text explaining the limitations of the investigation describe in the manuscript (in L57-64), which is relevant to understand the applicability of the presented results. This must be declared in the abstract in my opinion. In this sense, I do not think I am the right person to discuss this point, but I raise some doubts about the real scope and applicability of the manuscript for a high-impact factor journal as Nature Communications.

2) The authors demonstrate that significant differences are observed between surface chemistry of ash and bulk composition of pyroclasts, but they do not address if these differences can be effectively relevant when environmental interactions and eventual climate effects are quantified. I don't know if it's feasible, but it would strongly reinforce the introduction.

All in all, I recommend to accept the article with minor revisions, even though I raise some doubts about the real scope and applicability of the manuscript for a journal such as Nature Communications.

L45: I do not understand the order criterion for references. Is this (11,12) located before (9,10) ok?

L48: It is not clear how "chemistry" can create heterogeneity.

L50: toughness, or the resistance > toughness and/or resistance.

L62-63: I understand the inclusion of this limitation in the manuscript, but this implication is not clear to me from a logical point of view, even though it seems reasonable.

L67: approx. > ~.

L85: approximately 0.1-3 > approximately, ratios between 0.1 and 3.

L87-88: I suggest to add numerical indications of the magnitude of depletion in each case.

L116: A space is lacking between "EPMA" and "(Supplementary".

L216: There are two spaces between "the" and "set".

L262: I suggest to delete "distance".

L268: I suggest to use re-mobilization instead of collapse.

L274: "fed" > "triggered". This is because the erupted material includes magma stalling in the edifice.

L305: A space is lacking between C and (1s).

L374: XRF > XRF analysis.

L393-394: I suggest to use "times" instead of "x".

L602: This is the only line ending with a dot.

L633: 10 or 30 > 10 and 30.

L640-641: matrix glass > glass matrix.

Reviewer #4 (Remarks to the Author):

The authors have greatly improved the manuscript, and all my remarks and comments have been thoroughly addressed. I think that the manuscript can be published in its present form.

To maintain consistency between Figure 2 and Supplementary Figure 2, I suggest that the scale of the latter figure is also changed in 0.1 - 10 log scale. The caption should read "Ratio" and not "Ration".

REVIEWERS' COMMENTS

Reviewer #3 (Remarks to the Author):

Dear editor,

in this contribution, Hornby et al. address the influence of Fe-rich nanophases on fracture dynamics of pyroclasts during magmatic fragmentation and on the surface chemistry of the resulting ash, which is significantly different from the bulk composition of pyroclasts. The study is based on controlled fragmentation experiments on andesitic products, which derive from the 2006 sub-Plinian eruption of Tungurahua volcano. The paper is well-written and, in general, well-organized. The introduction is self-consistent in terms of references, description of the problem addressed in the text and explanation of the scope of the paper, while conclusions are effectively based on results. The authors addressed successfully the four major comments presented in my previous review, or at least they included additional discussions about some limitations of their results. I acknowledge particularly the modifications associated with my previous comments 2 and 4, related to the dynamics of magma fragmentation in controlled experiments and the eventual systematic discrepancies between pyroclasts compositional results derived from different measurement methods. I have two general, additional comments and a series of editorial comments and suggestions are presented at the end of this text (please note that my mother-tongue is not English.

We thank this reviewer for his continued close reading and interest in our work.

1) The authors included a text explaining the limitations of the investigation describe in the manuscript (in L57-64), which is relevant to understand the applicability of the presented results. This must be declared in the abstract in my opinion. In this sense, I do not think I am the right person to discuss this point, but I raise some doubts about the real scope and applicability of the manuscript for a high-impact factor journal as Nature Communications.

In the abstract, we describe the fracture-focusing model derived from our micro-to-nano textural and chemical study of the experimental materials, then state

In this manner, we argue that commonly-observed pre-eruptive microtextures can generate primary discrepancies in ash surface chemistry...

Given that fracture propagation during primary fragmentation of magma is a prerequisite for ash formation, we believe this revised statement effectively summarizes the implications that we expand on in the Introduction (lines 57-64). We also have reorganized the final few sentences of the Abstract to highlight the extent of comparable microtextures. The final sentence has been expanded to read:

In this manner, we argue that commonly-observed pre-eruptive microtextures caused by disequilibrium crystallization and/or unmixing can modify fracture propagation and generate primary discrepancies in ash surface chemistry...

2) The authors demonstrate that significant differences are observed between surface chemistry of ash and bulk composition of pyroclasts, but they do not address if these differences can be effectively relevant when environmental interactions and eventual climate effects are quantified. I don't know if it's feasible, but it would strongly reinforce the introduction.

This question highlights the exact importance of our work, and we appreciate the reviewer asking. While that is the ultimate implication of our findings, doing so in the Introduction would require extensive speculation of various impacted systems as well as poorly constrained extrapolation of ash properties. Instead, we discuss the range of systems potentially impacted (lines 44-46) in order to convey the wide applicability of our study within Nature word limits. We then, in lines 256-259, provide a qualitative example, stating

For example, emissions of Ca- and Fe-rich ash may be inferred from bulk chemistry to be a sink for volcanogenic SO₂^{1,47} and a contributor of bioavailable Fe to surface waters; however, a nanoscale surface enriched in alkalis and silica would likely be of little relevance to either phenomena.

As evidenced through previous rounds of review, how our findings translate to other grain sizes and eruption styles/compositions needs to be explored in detail before such an exercise would hold quantitative value. We'd also suggest that a modelling approach may be better suited for many case studies, where eruption source parameters could be varied to establish the extent of impacts on receiving systems. As outlined in the manuscript, our findings provide a path towards addressing the reviewer's question and we very much look forward to such work!

All in all, I recommend to accept the article with minor revisions, even though I raise some doubts about the real scope and applicability of the manuscript for a journal such as Nature Communications.

L45: I do not understand the order criterion for references. Is this (11,12) located before (9,10) ok?

Thank you, now corrected.

L48: It is not clear how "chemistry" can create heterogeneity.

True; this is now deleted and 'texture' changed to 'microtextures'.

L50: toughness, or the resistance > toughness and/or resistance.

The second part of this sentence was meant to be a description of fracture toughness – we have now included the statement in parentheses instead.

L62-63: I understand the inclusion of this limitation in the manuscript, but this implication is not clear to me from a logical point of view, even though it seems reasonable.

We revised slightly to improve clarity:

We do not argue for universal applicability of the specific fracture-focusing model outlined here to all volcanic surfaces, but rather use it to underpin an essential argument: if chemically discrepant ash surfaces are created from combinations of microtextural and fragmentation conditions in one situation, then there may be an array of variably discrepant surfaces created by interactions between microtexture and fracture propagation across the spectrum of ash-forming eruptive events.

L67: approx. > ~.

Done.

L85: approximately 0.1-3 > approximately, ratios between 0.1 and 3.

Changed as suggested

L87-88: I suggest to add numerical indications of the magnitude of depletion in each case.

Changed as suggested.

L116: A space is lacking between "EPMA" and "(Supplementary)".

Corrected.

L216: There are two spaces between "the" and "set".

Corrected.

L262: I suggest to delete "distance".

Changed as suggested.

L268: I suggest to use re-mobilization instead of collapse.

In this case we feel that collapse is a more specific description.

L274: "fed" > "triggered". This is because the erupted material includes magma stalling in the edifice.

Changed as suggested.

L305: A space is lacking between C and (1s).

Corrected.

L374: XRF > XRF analysis.

Corrected.

L393-394: I suggest to use "times" instead of "x".

Changed as suggested.

L602: This is the only line ending with a dot.

Corrected.

L633: 10 or 30 > 10 and 30.

Corrected.

L640-641: matrix glass > glass matrix.

Changed as suggested.

Reviewer #4 (Remarks to the Author):

The authors have greatly improved the manuscript, and all my remarks and comments have been thoroughly addressed. I think that the manuscript can be published in its present form.

To maintain consistency between Figure 2 and Supplementary Figure 2, I suggest that the scale of the latter figure is also changed in 0.1 - 10 log scale. The caption should read "Ratio" and not "Ration".

Thank you, both points have been corrected.